# Uncertainty Prioritized Experience Replay

## Abstract

Prioritized experience replay, which improves sample efficiency by selecting relevant transitions to update parameter estimates, is a crucial component of contemporary value-based deep reinforcement learning models. Typically, transitions are prioritized based on their temporal difference error. However, this approach is prone to favoring noisy transitions, even when the value estimation closely approximates the target mean. This phenomenon resembles the *noisy TV* problem postulated in the exploration literature, in which exploration-guided agents get stuck by mistaking noise for novelty. To mitigate the disruptive effects of noise in value estimation, we propose using epistemic uncertainty to guide the prioritization of transitions from the replay buffer. Epistemic uncertainty quantifies the uncertainty that can be reduced by learning, hence reducing transitions sampled from the buffer generated by unpredictable random processes. We first illustrate the benefits of epistemic uncertainty prioritized replay in two tabular toy models: a simple multi-arm bandit task, and a noisy gridworld. Subsequently, we evaluate our prioritization scheme on the Atari suite, outperforming quantile regression deep Q-learning benchmarks; thus forging a path for the use of epistemic uncertainty prioritized replay in reinforcement learning agents.

## 1 Introduction

Deep Reinforcement Learning (DRL) has proven highly effective across a diverse array of problems, consistently yielding state-of-the-art results in control of dynamical systems (Nian et al., 2020; Degrave et al., 2022; Weinberg et al., 2023), abstract strategy games (Mnih et al., 2015; Silver et al., 2016), continual learning (Khetarpal et al., 2022; Team et al., 2021), and multi-agent learning (OpenAI et al., 2019; Baker et al., 2020). It has also been established as a foundational theory for explaining phenomena in cognitive neuroscience (Botvinick et al., 2020; Subramanian et al., 2022). Nonetheless, a significant drawback of these methods pertains to their inherent *sample inefficiency* whereby accurate estimations of value and policy necessitate a substantial demand for interactions with the environment.

Sample inefficiency has been mitigated through the use of—among other methods—Prioritized Experience Replay (PER) (Schaul et al., 2016). PER is an extension of Experience Replay (Lin, 1992), which uses a memory buffer populated with past agent transitions to improve training stability through the temporal de-correlation of data used in parameter updates. Subsequently, PER extends this approach by sampling transitions from the buffer with probabilities proportional to their absolute Temporal Difference (TD) error, thereby allowing agents to *prioritise* learning from pertinent data. PER has been widely adopted as a standard technique in DRL; however, despite significantly better performance over uniform sampling in most cases, it is worth noting that PER can encounter limitations under specific task conditions and agent designs. The most prominent example of such a limitation is related to the so-called *noisy TV* problem (Burda et al., 2018), a thought experiment at the heart of the literature around exploration in RL. Just as novelty-based exploration bonuses can trap agents in noisy states, PER is susceptible to frequently replaying transitions involving high levels of randomness (e.g. in reward or transition dynamics) even if they do not translate to meaningful learning and thus are not useful for solving the task.

To combat this issue, we propose combining epistemic and aleatoric uncertainty measures (Clements et al., 2020; Alverio et al., 2022; Lahlou et al., 2022; Liu et al., 2023; Jiang et al., 2023), originally used to promote exploration, under an information gain criterion for use in replay prioritization. Epistemic uncertainty, the uncertainty reducible through learning, is the key quantity of interest.

However this need to be appropriately 'calibrated', which we show-both empirically, and with justification from Bayesian inference-can be done effectively by dividing the epistemic uncertainty estimate by an aleatoric uncertainty estimate (and taking the logarithm, i.e. the information gain). Intuitively the need for this kind of calibration can be seen by considering the following game: the aim is to estimate the mean of two distributions; the ground truth is that both distributions have identical mean but different variance, and your current estimates for both distributions are the same i.e. your epistemic uncertainty on the mean is the same for both distributions. However if I offer you a new sample from either distribution to refine your estimate you would choose to sample the distribution with lower variance since this is more likely to be informative. In addition to arguing for this novel prioritization variable, we also provide candidate methods involving distributions of ensembles (in the vein of Clements et al. (2020)) to estimate these quantities.

Our primary contributions are as follows: (1) In Section 3, we present a novel approach for estimating epistemic uncertainty, building upon an existing uncertainty formalisation introduced by Clements et al. (2020) & Jiang et al. (2023). This extension incorporates information about the target value that the model aims to estimate thereby accounting for bias in the estimator; (2) We derive a prioritisation variable using estimated uncertainty quantities, finding a specific functional form derived from a concept called *information gain*, showing that both, epistemic and aleatoric uncertainty should be considered for prioritisation; (3) In Section 4, we illustrate the advantages of our proposed epistemic uncertainty prioritisation scheme through two interpretable toy models—a bandit task and a grid world; (4) In Section 5, we demonstrate the effectiveness of this method on the Atari-57 benchmark (Bellemare et al., 2013), where it significantly outperforms baseline models based on a combination of PER, QR-DQN and ensemble agents.

## 2 BACKGROUND

### 2.1 REINFORCEMENT LEARNING

Consider an environment modelled by a Markov Decision Process (MDP), defined by $(S, A, R, P, \gamma)$ with state space $S$, action space $A$, reward function $R$, state-transition function $P$, and discount factor $\gamma \in (0, 1)$. Given the agent policy $\pi : S \to \Delta(A)$, where $\Delta(A)$ denotes the probability simplex over $A$, the cumulative discounted future reward is denoted by $G^\pi(s, a) = \sum_t \gamma^t R(s_t, a_t)$ with $s_0 = s$ and $a_0 = a$, and transitions sampled according to $a_t \sim \pi(a|s_t)$ and $s_{t+1}, r_t \sim P(s, r|s_t, a_t)$. We denote the action-value function as $Q^\pi(s, a) = \mathbb{E}[G^\pi(s, a)]$, and the corresponding state-action return-distribution function as $\eta^\pi(s, a)$; and we recall that $Q^\pi(s, a) = \mathbb{E}_{G \sim \eta^\pi(s, a)}[G]$. In general, the action value function is parameterized by $\psi$, such that $Q_\psi$ can be trained by minimizing a mean-squared temporal difference (TD) error $\mathbb{E}[\delta_t^2]$. For example, in Q-Learning the error is given by

$$\delta_t = r_t + \gamma \max_{a' \in A} Q_{\bar\psi}(s_{t+1}, a') - Q_\psi(s_t, a_t), \tag{1}$$

for the *transition* at time $t$ $(s_t, a_t, r_t, s_{t+1})$, and where $\bar\psi$ denotes the possibly time-lagged *target parameters* (Watkins & Dayan, 1992; Mnih et al., 2015). Additionally, we will use policies that are $\epsilon$-greedy with respect to the currently estimated action-value function, that is for some $\epsilon \in [0, 1]$, the selected action from any state $s$ is drawn as $\arg\max_{a \in A} Q_\psi(s, a)$ with probability $1 - \epsilon$ and uniformly over $A$ otherwise. See Sutton & Barto (2018) for a more in-depth overview of RL methods.

### 2.2 PRIORITIZED EXPERIENCE REPLAY

Reinforcement learning algorithms are notoriously sample inefficient. A widely adopted practice to mitigate this issue is the use of an experience replay buffer, which stores transitions in the form of $(s_t, a_t, r_t, s_{t+1})$ for later learning (Mnih et al., 2015). Loosely inspired by hippocampal replay to the cortex in mammalian brains (Foster & Wilson, 2006; McNamara et al., 2014), its primary conceptual motivation is to reduce the variance of gradient-based optimization by temporally de-correlating updates, thereby improving sample efficiency. It can also serve to prevent catastrophic forgetting by maintaining transitions from different time scales. The effectiveness of this buffer can often be improved further by *prioritising* some transitions at the point of sampling rather than selecting uniformly. Formally, when transition $i$ is placed into replay, it is given a priority $p_i$. The probability

of sampling this transition during training is given by:

$$P(i) = \frac{p_i^\alpha}{\sum_k p_k^\alpha}, \tag{2}$$

where $\alpha$ is a hyper-parameter called *prioritisation exponent* ($\alpha = 0$ corresponds to uniform sampling). Schaul et al. (2016) introduced *prioritized experience replay*, which most often uses the absolute TD-error $|\delta_i|$ of transition $i$, as $p_i = |\delta_i| + \epsilon$ where a small $\epsilon$ constant ensures transitions with zero error still have a chance of being sampled[1]. Sampling transitions non-uniformly from the replay buffer will change the observed distribution of transitions, biasing the solution of value estimates. To correct this bias, the error used for each update is re-weighted by an importance weight of the form $w_i \propto (NP(i))^{-\beta}$, where $N$ is the size of the buffer and $\beta$ controls the correction of bias introduced by important sampling ($\beta = 1$ corresponds to a full correction).

The key intuition behind PER is that transitions on which the agent previously made inaccurate predictions should be replayed more often than transitions on which the agent already has low error. While this heuristic is reasonable and has enjoyed empirical success, TD-errors can be insufficiently distinct from the irreducible aleatoric uncertainty; considering instead uncertainty measures more explicitly, this form of prioritisation can be significantly improved.

### 2.3 UNCERTAINTY ESTIMATION IN RL

Uncertainty is a fundamental concept in statistics. Within machine learning, it has predominately been studied in supervised learning, particularly with Bayesian methods (Lahlou et al., 2022; Narimatsu et al., 2023). Various aspects of the task setting such as bootstrapping and non-stationarity make uncertainty estimation a significantly more challenging problem in RL; nevertheless, it has featured more prominently in recent work, including for use in generalization (Jiang et al., 2023), as reward bonuses in exploration (Nikolov et al., 2019), and to guide safe actions (Lütjens et al., 2019; Kahn et al., 2017). We discuss here some of the key concepts around uncertainty relevant to this work, particularly those that address the delineation between aleatoric and epistemic uncertainty. A more comprehensive overview of related work around uncertainty in RL can be found in Appendix A.

#### 2.3.1 BOOTSTRAPPED DQN

The concept behind bootstrapping is to approximate a posterior distribution by sampling a prediction from an ensemble of estimators, where each estimator is initialized randomly and observes a distinct subset of the data (Tibshirani, 1994; Bickel & Freedman, 1981). In RL, Osband et al. (2016) introduced a protocol known as *bootstrapped DQN* for deep exploration, whereby bootstrapping is used to approximate the posterior of the action-value function, from which samples can be drawn. Each agent within an effective ensemble, parameterized by $\psi$, is randomly initialized and trained using a different subset of experiences via random masking. A sample estimate of the posterior distribution, denoted as $\psi \sim P(\psi|D)$ ($D$ being training data), is obtained by randomly selecting one of the agents from the ensemble. In this work, we use extensions of the bootstrapped DQN idea in our epistemic uncertainty measurements—notably the ensemble disagreement.

#### 2.3.2 DISTRIBUTIONAL RL

Learning quantities beyond the mean return has been a long-standing programme of RL research, with particular focus on the return variance (Sobel, 1982). A yet richer representation of the return is sought by more recent methods known collectively as *distributional RL* (Bellemare et al., 2017), which aims to learn not just the mean and variance, but the entire return distribution. We focus here on one particular class of distributional RL methods: those that model the quantiles of the distribution, specifically QR-DQN (Dabney et al., 2017). A broader treatment of the distributional RL literature can be found in Bellemare et al. (2023).

In QR-DQN, the distribution of returns, for example from taking action $a$ in state $s$ and subsequently following policy $\pi$, $\eta^\pi(s, a)$ is approximated as a *quantile representation* (Bellemare et al., 2023), that is, as a uniform mixture of Diracs, and trained through *quantile regression* (Koenker & Hallock, 2001).

---

[1]Another form of prioritization, known as rank-based prioritisation, is to use $p_i = 1/\text{rank}(i)$ where $\text{rank}(i)$ is the rank of the experience in the buffer when ordered by $|\delta_i|$.

For such a distribution, $\hat{\nu} = \frac{1}{m}\sum_{i=1}^{m}\delta_{\theta_{\tau_i}}$, with learnable quantile values $\theta_{\tau_i}$ and corresponding quantile targets $\tau_i = \frac{2i-1}{2m}$, the quantile regression loss for target distribution $\nu$ is given by

$$\mathcal{L}_{\text{QR}} = \sum_{i=1}^{m} \mathbb{E}_{Z\sim\nu}[\rho_{\tau_i}(Z - \theta_{\tau_i})], \tag{3}$$

where $\rho_\tau(u) = u(\tau - \mathbb{1}_{u<0})$ and $\mathbb{1}$ is the indicator function. By leveraging the so-called distributional Bellman operator and the standard apparatus of a DQN model, QR-DQN prescribes a temporal difference deep learning method for minimising the above loss function and learning an approximate return distribution function via quantile regression.

Distributional RL in itself does not (so far) permit a natural decomposition of uncertainties into epistemic and aleatoric (Clements et al., 2020; Chua et al., 2018; Charpentier et al., 2022); rather the variance of the learned distribution will converge on what can reasonably be thought of as the aleatoric uncertainty. In Subsection 3.1 we extend previous techniques that combine distributions with ensembles to construct estimates of both epistemic and aleatoric uncertainties. Both of these techniques to characterise epistemic uncertainty can be understood under an excess risk framework, which we outline below.

### 2.3.3 DIRECT EPISTEMIC UNCERTAINTY PREDICTION

We employ a clear and formal representation of uncertainty, where total uncertainty is defined as the sum of epistemic and aleatoric components such that the epistemic uncertainty can be interpreted as the excess risk. This notion was introduced by Xu & Raginsky (2022) and later extended by Lahlou et al. (2022); we adapt their framing to our setting here. Consider the **total uncertainty** $\mathcal{U}(s, a)$ of an action-value predictor $Q_\psi(s, a)$, for a given state $s$ and action $a$ as:

$$\mathcal{U}(Q_\psi, s, a) = \int \left(\Theta(s', r) - Q_\psi(s, a)\right)^2 P(s', r|s, a)ds'dr, \tag{4}$$

where $\Theta(s', r)$ is the Q-learning target as in equation 1. Then, the **aleatoric uncertainty** $\mathcal{A}(s, a)$, is given by the total uncertainty (as defined above) of a Bayes-optimal predictor $Q_\psi^*$ (see Lahlou et al. (2022)):

$$\mathcal{A}(s, a) = \mathcal{U}(Q_\psi^*, s, a). \tag{5}$$

Note that this quantity is independent of any learned predictor and is a function of the data only. The **epistemic uncertainty** $\mathcal{E}(Q_\psi, s, a)$, which is computed for a given predictor, is defined as the total uncertainty of the predictor minus the aleatoric uncertainty:

$$\mathcal{E}(Q_\psi, s, a) = \mathcal{U}(Q_\psi, s, a) - \mathcal{A}(s, a), \tag{6}$$

where $\mathcal{E}(Q_\psi, s, a)$ is the squared distance between the true mean and estimate mean as shown in Appendix C. Concretely, this decomposition can be useful in instances where you want to estimate epistemic uncertainty, but doing so directly is significantly more difficult than estimating total and aleatoric uncertainty, which is often the case. In Section 3, we provide a way to estimate quantities in this manner, which later we use to prioritise transitions in the replay buffer.

### 2.3.4 ENSEMBLES OF DISTRIBUTIONS

Using an ensemble of distributional RL agents gives us a concrete prescription for computing epistemic uncertainty as well as aleatoric uncertainty. This approach was first formalised by Clements et al. (2020), who define learned aleatoric and epistemic uncertainty quantities as a decomposition of the variance of the estimation from the ensemble (here defined as total uncertainty $\hat{\mathcal{U}}$) of distributional RL agents:

$$\hat{\mathcal{U}}(s, a) = \mathbb{V}_{\tau,\psi}\left[\theta_\tau(s, a; \psi)\right] = \hat{\mathcal{E}}(s, a) + \hat{\mathcal{A}}(s, a) \tag{7}$$

where

$$\hat{\mathcal{A}}(s, a) = \mathbb{V}_\tau[\mathbb{E}_\psi(\theta_\tau(s, a; \psi))], \quad \hat{\mathcal{E}}(s, a) = \mathbb{E}_\tau[\mathbb{V}_\psi(\theta_\tau(s, a; \psi))], \tag{8}$$

and $s$, $a$ are state and action, $\psi \sim P(\psi|D)$ are the model parameters of each agent in the ensemble, $D$ denotes the data distribution, and $\theta_\tau$ is the value of the $\tau^{\text{th}}$ quantile. $\mathbb{V}$ and $\mathbb{E}$ are variance and expectation operators respectively. Intuitively, $\hat{\mathcal{E}}$ measures epistemic uncertainty as the expected disagreement (variance) in quantile estimations across the ensemble, while $\hat{\mathcal{A}}$ takes the average estimation across the ensemble for each quantile of the distribution, and computes the variance of this averaged distribution. Clements et al. (2020) stop short of using a *bona fide* ensemble to estimate these quantities, opting instead for a two-sample approximation in the agent they present. However Jiang et al. (2023) go on to use ensemble methods more explicitly, as we do in this work.

## 3 UNCERTAINTY PRIORITISED EXPERIENCE REPLAY

In this section we will introduce a new method for estimating epistemic uncertainty, which arises from a decomposition of the total uncertainty as defined by the average error over both the ensemble and quantiles. This decomposition is in the vein of Clements et al. (2020); however, it considers distance from the target in addition to the disagreement within the ensemble, thereby allowing us to handle—among others—model bias. We go on to derive an expression for prioritisation variables based on the concept of *information gain*, which trades off epistemic and aleatoric uncertainty with a view to maximizing learnability from each sampled transition. We name this method Uncertainty Prioritised Experience Replay (UPER). Importantly, we are not changing the prioritize replay algorithm itself, but just the variable $p_i$ used to prioritise in Equation 2, replacing the TD-error by the information gain.

### 3.1 UNCERTAINTY FROM DISTRIBUTIONAL ENSEMBLES

The definitions given in Equation 7 arise from a decomposition of $\mathbb{V}_{\psi,\tau}[\theta_\tau(s, a; \psi)]$, where $\psi$ and $\tau$ index the quantile and ensemble respectively (see Clements et al. (2020) for details). This quantity does not explicitly consider how far estimates are from targets, but rather how consistent the estimates are among the quantiles and members of the ensemble. We propose a modified concept of total uncertainty $\hat{\mathcal{U}}_\delta$ named *target total uncertainty*, simply defined as the average squared error to the target $\Theta$ over the quantiles and ensemble, which can be decomposed as:

$$\hat{\mathcal{U}}_\delta = \mathbb{E}_{\tau,\psi}[(\Theta(s', r) - \theta_\tau(s, a; \psi))^2] = \underbrace{\delta_\Theta^2(s, a) + \hat{\mathcal{E}}(s, a)}_{\hat{\mathcal{E}}_\delta(s,a)} + \hat{\mathcal{A}}(s, a); \qquad (9)$$

where $\delta_\Theta^2(s, a) = (\Theta(s', r) - \mathbb{E}_{\tau,\psi}[\theta_\tau(s, a; \psi)])^2$, and we introduce the *target epistemic uncertainty* $\hat{\mathcal{E}}_\delta(s, a) = \delta_\Theta^2(s, a) + \hat{\mathcal{E}}(s, a)$ (see proof of this decomposition in Appendix B). Note that in order to construct ensemble disagreement estimates or estimates of the total uncertainty $\hat{\mathcal{U}}_\delta$, we assume independence among the ensemble, which is facilitated by masking and random initialisation akin to bootstrapped DQN. Through the lens of the DEUP formulation from Section 2.3.3, this decomposition suggests a modified definition of epistemic uncertainty that considers the distance to the target $\delta_\Theta^2$ as well as the disagreement in estimation within the ensemble $\hat{\mathcal{E}}$ from Clements et al. (2020) and Jiang et al. (2023). To see why this extra term can be useful, consider the following pathological example: all members of an ensemble are initialised equally, the variance among the ensemble—and the resulting epistemic uncertainty estimate without this additional error term—will be zero. A more subtle generalisation of this would be if inductive biases from other parts of the learning setup (architecture, learning rule etc.) lead to characteristic learning trajectories in which individual members of the ensemble effectively collapse with no variance. In essence, $\hat{\mathcal{E}}$ assesses ensemble disagreement without including the estimation offset. The use of pseudo-counts (Lobel et al., 2023) presents a similar problem: while epistemic uncertainty does scale with the number of visits to a state, it does not necessarily encode the true distance between the estimation and target values. Pseudo-counts bear the additional disadvantage of being task agnostic, i.e. ignoring context, which makes them brittle under any change in the underlying MDP. We provide a simulation where we show the advantage of using $\hat{\mathcal{E}}_\delta$ instead of $\hat{\mathcal{E}}$ to prioritise replay in Section 4.

## 3.2 PRIORITISING USING INFORMATION GAIN

Having arrived at suitable methods for estimating both epistemic and aleatoric uncertainty, it remains to establish a functional form for the prioritisation variable, denoted $p_i = h(\mathcal{E}(s_i, a_i), \mathcal{A}(s_i, a_i))$. The most straightforward approach is to directly use $p_i = \hat{\mathcal{E}}_\delta$; however, in practical applications, this does not yield satisfactory results. One intuition for this, which will be made more concrete in later passages, is that the magnitude of epistemic uncertainty does not in itself determine how easily reducible that uncertainty is. It is informative therefore to also consider the aleatoric uncertainty, since this indicates the fidelity of the data, and hence how readily it can be used to reduce the epistemic uncertainty (this is demonstrated experimentally in Subsection 4.1 and Appendix E, and expounded upon in Appendix D).

We take inspiration from the idea of *information gain* to determine $h$. For the purpose of this explanation, consider a hypothetical dataset of points $x_i \sim \mathcal{N}(\mu_x, \sigma_x^2)$. Our objective is to estimate the posterior distribution $P(\nu|x_i) \propto P(x_i|\nu)P(\nu)$ with a prior distribution $\nu \sim \mathcal{N}(\mu, \sigma^2)$. Following the observation of a single sample $x_i$, the posterior distribution becomes a Gaussian with variance $\sigma_\nu^2 = \frac{\sigma^2 \sigma_x^2}{\sigma_x^2 + \sigma^2}$. To quantify the information gained by incorporating the sample $x_i$ when computing the posterior, we measure the difference in entropy between the prior distribution and the posterior as

$$\Delta \mathcal{H} = \mathcal{H}\left(P(\nu)\right) - \mathcal{H}\left(P(\nu|x_i)\right), \tag{10}$$

From here, we consider $\sigma^2 = \hat{\mathcal{E}}_\delta$ as a form of epistemic uncertainty, since the ensemble disagreement is reduced by sampling more points, and $\sigma_x^2 = \hat{\mathcal{A}}$ as aleatoric uncertainty corresponding to the variance of the ensemble average distribution, giving the irreducible noise of the data, obtaining a prioritisation variable

$$p_i = \Delta \mathcal{H}_\delta = \frac{1}{2} \log \left(1 + \frac{\hat{\mathcal{E}}_\delta(s, a)}{\hat{\mathcal{A}}(s, a)}\right). \tag{11}$$

For a detailed derivation of the information gain, an illustrative simulation demonstrating the use of variance as an uncertainty estimate, and a comprehensive exploration of other functional forms of prioritization variables based on uncertainty, please refer to Appendix D.

## 4 MOTIVATING EXAMPLES

We proceed to employ epistemic uncertainty estimators and the information gain criterion in simple and interpretable toy models to highlight their potential as experience replay prioritisation variables.

### 4.1 CONAL BANDIT

We devise a multi-armed bandit task in which each arm has the same expected reward but with increasing noise level as per arm, forming a *cone* as shown from left to right in Figure 1a. The memory buffer in this experiment has one transition per arm, and after sampling one arm, the observed reward is replaced in the buffer for the respective transition (as done in the toy example in the original PER paper (Schaul et al., 2016)). Specifically, let $n_a$ denote the number of arms; then the reward distribution $r$ for arm $a$ is defined as:

$$r(a) = \bar{r} + \eta \cdot \sigma(a), \quad \sigma(a) = a \cdot \sigma_{\max}/(n_a - 1) + \sigma_{\min}; \tag{12}$$

where $\bar{r}$ represents the expected reward, $\sigma(a)$ is the reward standard deviation associated with arm $a$, $\sigma_{\max}$ and $\sigma_{\min}$ are constant, and $\eta$ is sampled from a centred, unit-variance Gaussian.

The choice of employing noisy arms serves the purpose of demonstrating that the TD-errors will inherently include the sample noise, regardless of whether the reward estimation for each arm $Q(a) = \mathbb{E}_j[\theta_j(a)]$ approximates the target value $\bar{r}$. We depict results for the bandit task using different variables to prioritise learning in Figure 1b for $n_a = 5$, $\bar{r} = 2$, $\sigma_{\max} = 2$ and $\sigma_{\min} = 0.1$ (details in Appendix E).

Four relevant prioritisation schemes are shown in this section (see Appendix E for other prioritisation schemes): TD-error (standard PER): $p_i = \frac{1}{N_e} \sum_\psi |r_i - Q(a_i; \psi)|$; Inverse count: $p_i = 1/\sqrt{1 + C}$,

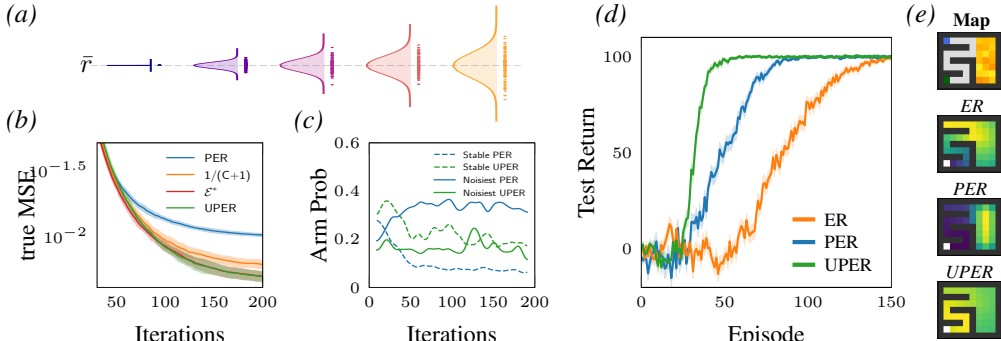

Figure 1: **Conal Bandit.** *(a)* multi-armed bandit task constructed such that each arm has identical mean payoff but increasing variance. *(b)* true MSE (average error across arms, between estimated reward and the true reward mean) over 200 iterations (each of 1000 steps) using different quantities to prioritise transitions from the replay buffer: absolute value of the TD error $|\delta|$ (PER), inverse counts ($C$ being the number of visits to the respective arm), information gain $\Delta\mathcal{H}_\delta$ (UPER), and an oracle epistemic uncertainty $\mathcal{E}^*$ measured as the distance from the estimated mean to the true mean. *(c)* arm replaying selection probabilities for the stablest (dashed) and noisiest (solid) arms in the conal bandit; the key intuition is that prioritising by TD-error over-samples noisier arms, while prioritising using UPER places importance on learn-ability and leads to greater selection of stable arms. Results averaged across 10 seeds. **Noisy Gridworld.** *(d)* 300 seeds return on a test episode throughout training of an agent on the noisy gridworld, with the shaded region being stared error on the mean. *(e)* in the **Map**, blue denotes the starting state, green is the goal state, and yellow are the non-zero variance immediate rewards. Below, sampling heatmaps where yellows are highly sampled and blues are scarcely sampled: uniform experience replay (ER) leads to sampling more from early parts of a trajectory since these fill the buffer first; replay based on TD error (PER) leads to a pathological sampling of the noisy part of the gridworld; replay using UPER leads to greater sampling of later parts of the trajectory.

where $C$ denotes the number of times an arm has been sampled to update the reward estimate; Information gain (UPER): $p_i = \Delta\mathcal{H}_\delta$; True distance to target: $p_i = \mathcal{E}^* = |\bar{r} - Q(a_i)|$.

Prioritizing with epistemic uncertainty measures, such as UPER or inverse counts (a proxy for epistemic uncertainty), leads to improved training speed and final true Mean Squared Error (true MSE, averaged across all arms, between the estimated reward and the true mean reward), compared to $p_i = |\delta_i|$ (PER), as illustrated in Figure 1b. Throughout the paper, we highlight that the TD-error includes aleatoric uncertainty, corresponding to the arm variance in this scenario—which is irreducible through learning (see Subsection C.1 for more details). Therefore, the TD-error tends to over-sample arms with high variance compared with UPER, to the cost of not sampling the low variance arm. This is demonstrated in Figure 1c.

Using inverse counts as the prioritization variable (similar to Lobel et al. (2023)), outperforms TD-error (as designed in the task) but not UPER. The reason is the fact that, although each initial estimated Q-values per arm are equidistant from the true mean, the learning speed for each arm diminishes with the variance of the respective arm. Inverse counts do not account for this variance-dependent decay in learning speed, so the number of updates per arm will not reflect the distance of the estimation to the true target, whereas UPER (prioritising by $\hat{\mathcal{E}}_\delta$ and inverse $\hat{A}$) tends to sample arms with high aleatoric uncertainty less frequently, and is also based on the distance to the targets as defined in Equation 9.

The distance between the estimated mean and the true mean, denoted as $\mathcal{E}^*$ (accessible due to the task design), is equivalent to the epistemic uncertainty in the DEUP formulation, as derived in Appendix C. This distance is the ideal prioritisation variable to which we do not have access in general. Notably, using UPER, which prioritizes based on information gain, yields results comparable to prioritising directly based on the true distance. These results show UPER as a promising modification to TD-error-based prioritised replay.

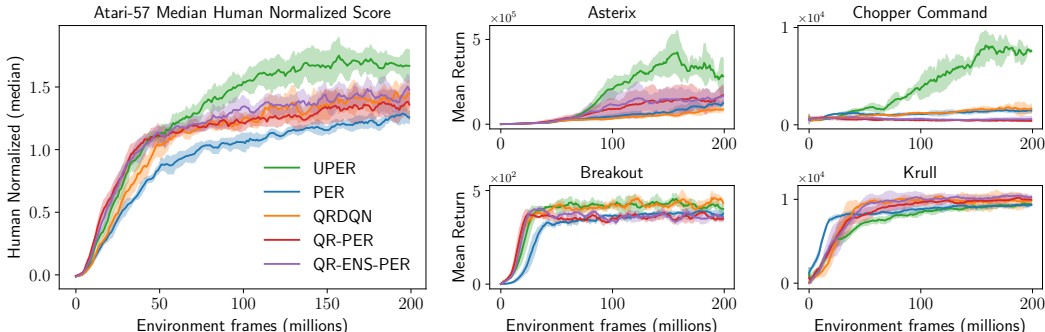

Figure 2: (**Left**) Comparing Uncertainty Prioritized Experience Replay (UPER) with Prioritized Experience Replay (PER) and QR-DQN on the full Atari-57 benchmark. Median human normalized score for UPER is significantly higher than baselines throughout the learning trajectory. (**Right**) Example of per-game performance, with vastly superior performance on e.g. Asterix and Chopper Command; cases in which UPER is worse are far less extreme, for instance Breakout and Krull (this is shown graphically in Figure 14 and Figure 15). All results are averages over 3 seeds.

To emphasise the significance of incorporating the target value when utilising the target epistemic uncertainty $\hat{\mathcal{E}}_\delta$ for replay prioritisation, we introduced modifications to the conal bandit task by assigning distinct mean rewards per arm, denoted as $\bar{r} \to \bar{r}(a)$ (see simulation details in Appendix E, Figure 6). In the original conal bandit task, all arms shared the same mean reward $\bar{r}$, resulting in an equal initial distance expectation from $Q(a)$ to each arm. This uniformity dampened the performance improvement when considering the target distance $\delta_\Theta$ in $\hat{\mathcal{E}}_\delta$ with respect to $\hat{\mathcal{E}}$. By introducing varying mean rewards per arm, denoted as $r(a)$, the relevance of information about the target value becomes important. This adjustment highlights the advantage of employing our proposed target epistemic uncertainty $\hat{\mathcal{E}}_\delta$ over merely considering ensemble disagreement $\hat{\mathcal{E}}$.

## 4.2 NOISY GRIDWORLD

In order to move toward the full RL problem, we consider in this section a tabular gridworld. We take inspiration from ideas in planning within dynamic programming methods (Moore & Atkeson, 1993) to probe uncertainty-guided prioritised replay. Typically under this framework, 'direct' reinforcement learning on interactions with the environment (sometimes referred to as control) is supplemented with 'indirect' learning of a model from stored experiences (sometimes referred to as planning). In our case, we learn purely model-free but retain these ideas of offline vs. online learning. In some ways these methods are the pre-cursor to the use of experience replay buffers in DRL. When making updates on stored data offline (for planning or otherwise), the same questions around criteria for prioritisation arise. Notably, prioritised sweeping (preference over high error samples in memory) was an early extension to the Dyna models that exemplify this learning protocol (Sutton, 1991). In Figure 1e Map, we construct a gridworld where the agent can encounter a set of very noisy states with random rewards early on in the episode while a single deterministic state with a much larger reward is at the end of the maze. Figure 1d shows that this simple task can be solved without the additional planning steps, but ER (sampling uniformly) helps improve sample efficiency. This is improved further by PER (prioritising using TD), but even more so by UPER where we prioritize using the information gain criterion and the inverse of state visitation counts (a good proxy for epistemic uncertainty in this tabular setting). As shown by the heatmaps in Figure 1e, PER over-samples noisy states while UPER prioritises on novel states towards the end of the trajectory. Full details of the experimental setup and hyper-parameters can be found in Appendix F.

## 5 DEEP RL: ATARI

In our final set of experiments we apply our insights in a DRL setting, specifically the Atari benchmark (Bellemare et al., 2013). Our agent is an ensemble of QR-DQN distributional predictors (N=10), in which experience replay is prioritized using the information gain (UPER in Subsection 3.2). We

compare this method to a vanilla QR-DQN agent (Dabney et al., 2017) with uniform prioritisation and the original PER agent (Schaul et al., 2016). To show that the gain in performance is not due to either the quantile regression method, nor the ensemble, we trained a QR-DQN agent with TD-error prioritization (QR-PER), and an ensemble of QR agents with TD-error prioritization (QR-ENS-PER). A summary of our empirical results is shown in Figure 2, with further ablations and details in Appendix G.

Except for the additional hyper-parameters associated with the ensemble of distributional prediction heads and a more commonly used configuration for the Adam optimizer ($\epsilon = 0.01/(\text{batch\_size})^2$), the network architecture and all hyper-parameters in UPER are identical to QR-DQN (Dabney et al., 2017). Likewise PER, QRDQN, and QR-PER baselines follow the implementations of Dabney et al. (2017) and Schaul et al. (2016) respectively, while QR-ENS-PER is identical to UPER except for the prioritisation variable which is TD-error. Concretely for the UPER agent, we compute the target epistemic uncertainty using $\hat{\mathcal{E}}_\delta(s, a) = \hat{\mathcal{U}}_\delta(s, a) - \hat{\mathcal{A}}(s, a)$. Then for a given transition $i$ the total uncertainty is given by

$$\hat{\mathcal{U}}_\delta = \mathbb{E}_{\tau, \tau', \psi} \left[ \left( r_i + \gamma \theta_{\tau'}(s_i', a_i'; \bar{\psi}) - \theta_\tau(s_i, a_i; \psi) \right)^2 \right], \tag{13}$$

where $\tau$ ($\tau'$) are the quantiles of the online (target) network $\psi$ ($\bar{\psi}$). The aleatoric uncertainty estimate is given by $\hat{\mathcal{A}}(s, a)$ in Equation 8. From these estimates we construct UPER priority variable using the uncertainty ratio discussed in Subsection 3.2, i.e. Equation 11. Since UPER and QR-ENS-PER are ensemble agents, we store a random mask $m \in \mathbb{R}^N$ for each transition in the buffer where $m_i \sim \mathcal{B}(0.5)$. When the transition is sampled for learning, gradients are only propagated for heads whose corresponding element in the mask is 1. This follows the proposal of (Osband et al., 2016) and serves to de-correlate the learning trajectories of the ensemble members, which is integral to the validity of our uncertainty estimates.

As depicted in Figure 2, the median UPER performance across games is significantly better than other prioritization schemes, showing that the performance improvement is not due to either the quantile regression technique or the ensemble alone. Importantly, UPER demonstrates performance improvement compared to its closest comparison QR-ENS-PER, whose only difference with UPER is the prioritization using TD-error (see Figure 14). In most games where UPER does not improve performance, such as Krull, Q*bert or H.E.R.O., the difference in performance is not significant. This is shown in the panels per game in Figure 15 and the assymetry of the bar plots in Appendix G.

## 6 RELATED WORK

**Exploration.** While UPER is not explicitly promoting exploration through a reward bonus to unexplored or uncertainty states, we borrow methods from this field to estimate epistemic and aleatoric uncertainty (Clements et al., 2020) to prioritize transitions from the replay buffer based on the information gain. A fundamental dilemma faced by RL agents is the exploration-exploitation trade-off (Osband et al., 2016; O'Donoghue, 2023), in which agents must balance competing objectives for action selection, between uncovering new information about the environment (exploration) and accumulating as much reward as they currently can (exploitation). Replay sampling and exploration strategies both affect the data used to enhance the estimation of the value function. The former controls the experiences used for value estimation updates, while the latter selects experiences that will end up populating the replay buffer. Many exploration strategies have been built around ideas of intrinsic reward (Oudeyer & Kaplan, 2007) and episodic memory (Savinov et al., 2019; Badia et al., 2020). These are susceptible to pathological behaviour induced by the noisy TV, and later variants are designed partly with this problem in mind; as a result they are frequently concerned with reliable and meaningful estimates of counts and novelty (Ostrovski et al., 2017b; Bellemare et al., 2016b; Burda et al., 2018; Lobel et al., 2023), dynamics (Stadie et al., 2015; Pathak et al., 2017), uncertainty (Mavor-Parker et al., 2022), and related quantities—many of which are relevant to our problem of constructing suitable measures for replay prioritization.

**PER.** Various efforts have been made to understand and improve upon aspects of prioritized experience replay since its introduction by Schaul et al. (2016). Integration of information related to uncertainty has often been in conjunction with strategies for managing the exploration-exploitation trade-off. For instance, in Sun et al. (2020), frequently visited states are sampled more frequently to reduce uncertainty around known states. Conversely, Alverio et al. (2022) approach is prioritizing

Table 1: Computational Cost (seconds per iteration)

| Architecture | CPU | GPU |
|---|---|---|
| QR-DQN-ENS | $28.40 \pm 0.26$ | $20.74 \pm 0.43$ |
| QR-DQN | $17.80 \pm 0.13$ | $18.49 \pm 0.68$ |
| DQN | $18.34 \pm 0.09$ | $18.39 \pm 0.56$ |

uncertain states to encourage exploration, utilizing epistemic uncertainty estimated as the standard deviation across an ensemble of next-state predictors. This technique is combined with other methods to enhance sample efficiency.

Another method, presented in Lobel et al. (2023), employs a pseudo-count approximation to gauge state visits, fostering exploration as an intrinsic reward. In training the pseudo-count network they prioritize transitions according to the counts themselves; they do not however go as far as performing this prioritisation for learning the actual value network—as is the focus of our work. The method of Lobel et al. (2023) allows estimation of epistemic uncertainty independent of the sparsity or density of the reward signal, making it especially appealing in sparse-reward environments. However, using pseudo-counts for epistemic uncertainty can also be poorly aligned with uncertainty about the actual value estimation problem (Osband et al., 2018). As described in Subsection 4.1, the number of visits to a specific state-action does not necessarily describe the error between the mean estimates to the true one. In addition to this, as explained in Subsection 3.2 and shown by simulation in Subsection 4.1, both epistemic and aleatoric uncertainty should be considered to build a proper prioritisation scheme.

## 7 DISCUSSION AND CONCLUSIONS

In this study, we propose using epistemic uncertainty measures to guide the prioritization of transitions from the replay buffer. We demonstrate both via mathematical analysis and careful experiments that the typically applied TD-error criterion can include aleatoric uncertainty, and lead to over-sampling of noisy transitions. Prioritizing by a principled function of epistemic and aleatoric uncertainty in the form of the information gain mitigates these effects. To construct this function, we expand the concept of epistemic uncertainty from Clements et al. (2020) to incorporate the distance to the target, achieving performance advantages in toy settings and complex problems such as the Atari 57 benchmark. In estimating these auxiliary quantities, one concern may be the increased computational cost in the deep learning setting. However, sharing of the lower level representation over multiple heads alongside efficient implementations can significantly mitigate this burden. To demonstrate this, we conducted an experiment on a lower-capacity GPU comparing the training times of DQN, QR-DQN, and QR-DQN + ensemble networks in the Pong environment. The time per iteration is presented in Table 1. The comparable training times can be attributed to effective batch processing facilitated by GPU parallelization. In our implementation, each agent in the ensemble is represented by a distinct output head in the network architecture. By extending the batch dimension to (batch, action, quantiles, ensemble), we leverage the parallelization capacity of the GPU, which still operates within capacity for the QR-DQN ensemble network. Further details of this experiment and the computer architecture used are presented in Subsection G.2. Note that this analysis does not aim to evaluate or compare the computational cost of sampling with a priority variable vs. uniform sampling. This is already addressed in the original PER paper and has negligible impact.

While we focus our implementation on distributional RL—a widely used set of methods, exploring other forms of uncertainty estimation in RL such as pseudo-counts (Lobel et al., 2023), in combination with different functional forms outside information gain, is a promising research path both for different prioritisation schemes and related parts of the RL problem like exploration (see Appendix D and Appendix E).

The framework of combining epistemic and aleatoric uncertainties in an information gain introduced in this work is not restricted to reinforcement learning. In principle, these concepts can be extrapolated to other learning systems. A substantial body of literature exists on the efficient selection of datapoints to enhance learning in other paradigms such as supervised (Hüllermeier & Waegeman, 2021; Zhou et al., 2022), continual (Henning et al., 2021; Li et al., 2021), or active learning (Nguyen et al., 2022). In addition, our work has the potential to offer alternative insights into replay events in biological agents (Daw et al., 2005; Mattar & Daw, 2018; Liu et al., 2019; Antonov et al., 2022).

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

## A   FURTHER RELATED WORK

In the main text we focus primarily on related work in uncertainty estimation for reinforcement learning that is specific to the epistemic vs. aleatoric dichotomy. Here we give an extended discussion on uncertainty estimation methods more generally.

### A.1   DIRECT VARIANCE ESTIMATION

Distributional RL provides a framework for computing statistics of the return beyond the mean. Efforts to compute such quantities in RL date back to Sobel (1982), who derived Bellman-like operators for higher order moments of the return in MDPs that can be used to indirectly estimate variance. This has since been extended to a greater set of problem settings and models (Prashanth & Ghavamzadeh, 2016; Tamar et al., 2016; White & White, 2016). More recently methods have also been developed to directly estimate variance (Tamar et al., 2012); arguably the simplest such scheme for TD(0) learning is the following update rule for the action-value variance $\hat{\mathcal{A}}(s, a)$ at state $s, a$ (re-stated from Sherstan et al. (2018) for state and action):

$$\hat{\mathcal{A}}_{t+1}(s, a) \leftarrow \hat{\mathcal{A}}_t(s, a) + \bar{\alpha}\bar{\delta}_t, \tag{14}$$

where

$$\bar{\delta}_t \leftarrow \bar{r}_{t+1} + \bar{\gamma}_{t+1}\hat{\mathcal{A}}_t(s', a') - \hat{\mathcal{A}}_t(s, a), \tag{15}$$

$$\bar{r}_{t+1} \leftarrow \delta_t^2, \tag{16}$$

$$\bar{\gamma}_{t+1} \leftarrow \gamma_{t+1}^2; \tag{17}$$

$\delta_t$ is the temporal difference error of on the mean value estimate, and $\bar{\alpha}$ is the variance learning rate. $\bar{r}$ can be thought of as a 'meta' reward for the variance estimate. This update corresponds to simply regressing on the square of the mean estimate error in a standard regression problem (single state, no concept of discounting) like in the bandit experiments shown in Section 4. This form of estimating aleatoric uncertainty does not require quantile regression, but

### A.2   BAYESIAN METHODS

A more comphrehensive Bayesian approach to the reinforcement learning problem can be formulated via so-called Bayes-adaptive Markov decision processes (BAMDPs) (Martin, 1967), where an agent continuously updates a belief distribution over underlying Markov decision processes. Solutions to BAMDPs are Bayes' optimal in the sense that they optimally trade off exploration and exploitation to maximise expected return. However, in all but the smallest environments and settings, learning over this entire belief distribution is intractable (Brunskill, 2012; Asmuth & Littman, 2012).

Posterior sampling, which can be viewed as the analogue of Thompson sampling for MDPs, has been a popular method to approximate the full Bayesian posterior e.g. via ensembles (Osband et al., 2016) or dropout (Gal & Ghahramani, 2016); extensions include provision of pseudo priors (Osband et al., 2018; 2021). While these approaches have been successful in some settings, they have few guarantees. A different line of work includes using methods such as meta-learning to reason on and train the approximate posterior (Zintgraf et al., 2019; Humplik et al., 2019).

With regards to the discussions on epistemic and aleatoric uncertainty, the above methods can give the model access to a distribution over parameters that can be sampled and operated on (e.g. to calculate variance). They do not however—Bayes optimal or not—lead *per se* to a decomposition into epistemic and aleatoric uncertainty.

### A.3   COUNTS

Another category of methods that are frequently used in reinforcement learning and related paradigms like bandits is based around notions of counts e.g. of state visitation. Such counts can be used to construct intervals/bounds on confidence of learned quantities. This is the foundation of well established exploration methods in tabular settings called upper confidence bounds (Auer, 2002b;a). In function approximation settings, much of the focus has been on constructing accurate *pseudo* counts that incorporate state similarities (Bellemare et al., 2016a; Ostrovski et al., 2017a; Tang et al.,

2017). Despite the well demarcated distinction between count-based methods and those that address the Bayesian posterior above, with access to any mean-zero unit-variance distribution, an ensemble of mean-predictors of that distribution can be used to estimate pseudo-counts (Lobel et al., 2023). As a result, it is generally possible to convert a Bayesian posterior into pseudo-counts.

### A.4 MODEL-BASED

A set of methods that is further removed from those used in our work, but are often motivated by similar questions consists of learning a model of the environment. Downstream quantities like the prediction error of the environment model can be used as proxies for uncertainty or novelty e.g. for exploration bonuses. Much of this work falls under the domain of intrinsic motivation (Barto, 2013). Some of the methods in this area e.g. curiosity (Pathak et al., 2017) attempt implicitly to make the distinction between epistemic uncertainty and aleatoric uncertainty to avoid the noisy TV problem.

### A.5 BEYOND THE PRIORITISATION VARIABLE

Altering the prioritized experience replay is not confined to changing the prioritization variable. In Zha et al. (2019), the replay policy is adapted through gradient optimization. Balaji et al. (2020) introduces a regularization technique, enhancing continual learning by storing a compressed network activity version for replay. Additional methods encompass the utilization of sub-buffers storing transitions at multiple time scales (Kaplanis et al., 2020), replay for sparse rewards (Andrychowicz et al., 2017; Nair et al., 2018), and employing diverse sampling strategies (Pan et al., 2022). Further endeavors are aiming to understand the effects of PER in RL (Liu & Zou, 2017; Fedus et al., 2020).

## B TOTAL ERROR DECOMPOSITION

Following the same notation as in Section 3, the averaged square error to the target $\Theta$ over the quantiles and ensemble indexed by $j$ and $\psi$ respectively:

$$\mathbb{E}_{\psi,j}[(\Theta - \theta_j(\psi))^2] = \int_\psi \frac{1}{N} \sum_j^N (\Theta - \theta_j(\psi))^2 P(\psi|D)d\psi, \tag{18}$$

$$= \int_\psi \frac{1}{N} \sum_j^N \left[\Theta - \theta_j(\psi) \pm \mathbb{E}_\psi(\theta_j(\psi))\right]^2 P(\psi|D)d\psi, \tag{19}$$

$$= \int_\psi \frac{1}{N} \sum_j^N \left[(\Theta - \mathbb{E}_\psi(\theta_j(\psi)))^2 + (\mathbb{E}_\psi(\theta_j(\psi)) - \theta_j(\psi))^2 \right. \tag{20}$$

$$\left. +2\left(\Theta - \mathbb{E}_\psi(\theta_j(\psi))\right)\left(\mathbb{E}_\psi(\theta_j(\psi)) - \theta_j(\psi)\right)\right] P(\psi|D)d\psi, \tag{21}$$

$$= \int_\psi \frac{1}{N} \sum_j^N \left(\Theta - \mathbb{E}_\psi(\theta_j(\psi))\right)^2 P(\psi|D)d\psi \tag{22}$$

$$+ \underbrace{\frac{1}{N} \sum_j^N \int_\psi \left(\mathbb{E}_\psi(\theta_j(\psi)) - \theta_j(\psi)\right)^2 P(\psi|D)d\psi}_{\hat{\mathcal{E}} \text{ in equation 8}}, \tag{23}$$

and the term in equation 21 is zero when integrating over $\psi$. Finally, the term in 22 is

$$\int_\psi \frac{1}{N} \sum_j^N \left(\Theta - \mathbb{E}_\psi(\theta_j(\psi))\right)^2 P(\psi|D)d\psi = \Theta^2 - 2\mathbb{E}_{\psi,j}\left(\theta_j(\psi)\right) + \mathbb{E}_j\left(\mathbb{E}_\psi\left[\theta_j(\psi)\right]^2\right) \tag{24}$$

$$= \underbrace{\left(\Theta - \mathbb{E}_{\psi,j}\left[\theta_j(\psi)\right]\right)^2}_{\text{Distance to the target } \delta_\Theta^2} + \underbrace{\mathbb{V}_j\left(\mathbb{E}_\psi\left[\theta_j(\psi)\right]\right)}_{\hat{\mathcal{A}} \text{ in equation 7}}. \tag{25}$$

Obtaining

$$\mathbb{E}_{\psi,j}[(\Theta - \theta_j(\psi))^2] = \delta_\Theta^2 + \hat{\mathcal{A}} + \hat{\mathcal{E}}. \tag{26}$$

## C   DEUP DECOMPOSITION

Consider the total uncertainty as defined in Lahlou et al. (2022) (but adapted for RL), which can be decomposed into epistemic uncertainty (distance between the mean estimation and true mean) and aleatoric uncertain (target variance) as:

$$\mathcal{U}(Q_\psi, s, a) = \int \left( \Theta(s', r) - Q_\psi(s, a) \right)^2 P(s', r | s, a) ds' dr \tag{27}$$

$$= \mathbb{E}_{s', r} \left[ \left( \Theta(s', r) - Q_\psi(s, a) \right)^2 \right] \tag{28}$$

$$= \mathbb{E}_{s', r} \left[ \Theta(s', r)^2 \right] - 2 Q_\psi(s, a) \mathbb{E}_{s', r} \left[ \Theta(s', r) \right] + Q_\psi(s, a)^2 \tag{29}$$

$$= \mathbb{V}_{s', r} \left[ \Theta(s', r) \right] + \mathbb{E}_{s', r} \left[ \Theta(s', r) \right]^2 - 2 Q_\psi(s, a) \mathbb{E}_{s', r} \left[ \Theta(s', r) \right] + Q_\psi(s, a)^2 \tag{30}$$

$$= \underbrace{\mathbb{V}_{s', r} \left[ \Theta(s', r) \right]}_{\text{aleatoric } \mathcal{A}(s,a)} + \underbrace{\left( Q_\psi(s, a) - \mathbb{E}_{s', r} \left[ \Theta(s', r) \right] \right)^2}_{\text{epistemic } \mathcal{E}(Q_\psi, s, a)} \tag{31}$$

### C.1   UNCERTAINTY DECOMPOSITION IN QUANTILE REGRESSION

Here we provide some extra intuition on the difference between MSE curves when prioritising by total uncertainty $\mathcal{U}$, td-error $|\delta|$, estimated epistemic uncertainty $\hat{\mathcal{E}}_\delta$ and true epistemic uncertainty $\mathcal{E}^*$. Let's start by considering a single agent trained using quantile regression as explained in Subsubsection 2.3.2. Consider the expected squared error of all quantiles indexed by $\tau$ and the target distribution $Z$, also defined in Subsection 3.1 as $\mathcal{U}$:

$$\mathcal{U}^2 = \mathbb{E}_{\tau, r \sim Z} \left[ (r - \theta_\tau)^2 \right] = \mathbb{E}_r \left[ r^2 \right] - 2 \mathbb{E}_r[r] \mathbb{E}_\tau \left[ \theta_\tau \right] + \mathbb{E}_\tau \left[ \theta_\tau^2 \right], \tag{32}$$

$$= \mathbb{V}_r \left[ r \right] + \bar{r}^2 - 2 \bar{r} Q(a) + Q(a)^2 + \mathbb{V}_\tau \left[ \theta_\tau \right], \tag{33}$$

$$= \underbrace{(\bar{r} - Q(a))^2}_{(\mathcal{E}^*)^2} + \underbrace{\mathbb{V}_r \left[ r \right]}_{\text{Target variance}} + \underbrace{\mathbb{V}_\tau \left[ \theta_\tau \right]}_{\text{Estimation variance}} . \tag{34}$$

The first term is the true epistemic uncertainty $\mathcal{E}^*$, second term and third term are the variance from the target, and the estimation variance. When using the total uncertainty as priority variable $p_i = \mathcal{U}$, the target and estimation uncertainty will be considered in the priority, therefore oversampling the noisiest arm as shown in the sampling probabilities depicted in Figures 7 and 8. When using the TD-error $p_i = |\delta_i|$, consider the expected squared TD-error

$$\mathbb{E}_r \left[ \delta^2 \right] = \left[ (r - \mathbb{E}_\tau \left[ \theta_\tau \right])^2 \right], \tag{35}$$

$$= \underbrace{(\bar{r} - Q(a))^2}_{(\mathcal{E}^*)^2} + \underbrace{\mathbb{V}_r \left[ r \right]}_{\text{Target variance}} . \tag{36}$$

Therefore, the TD-error does not prioritise by estimation variance, but it includes the target variance. Eventually, the target variance will be equal to the estimation variance, but from the start of the training, this is not true. Hence, the TD-error will also oversample the noisiest arm, but less compared to prioritising by total uncertainty $\mathcal{U}$. In practice, we do not have direct to $\mathbb{V}_r \left[ r \right]$, in fact this is a quantity we are trying to estimate by using quantile regression. We have implicit access to the true distance $\mathcal{E}^*$ (epistemic uncertainty) through the decomposition $\mathcal{U} = \mathcal{E} + \mathcal{A}$ as explain in Subsubsection 2.3.3, which is used to estimate epistemic uncertainty as in Section 3. Prioritising using information gain achieve similar results compare to the direct use of $\mathcal{E}^*$ to prioritise replay. For further discussion about epistemic uncertainty ratios, refer to Subsection D.3.

## D   PRIORITISATION QUANTITIES BASED ON UNCERTAINTY

### D.1   INFORMATION GAIN DERIVATION

Given the setup in Subsection 3.2, consider a hypothetical dataset of points $x_i \sim \mathcal{N}(\mu_x, \sigma_x^2)$. Our objective is to estimate the posterior distribution of the mean after observing one sample $P(\nu | x_i) \propto P(x_i | \nu) P(\nu)$ with a prior distribution of the mean $\nu \sim \mathcal{N}(\mu, \sigma^2)$. Following the observation of a single sample $x_i$, the posterior distribution is Gaussian with variance $\sigma_\nu^2 = \frac{\sigma^2 \sigma_x^2}{\sigma_x^2 + \sigma^2}$.

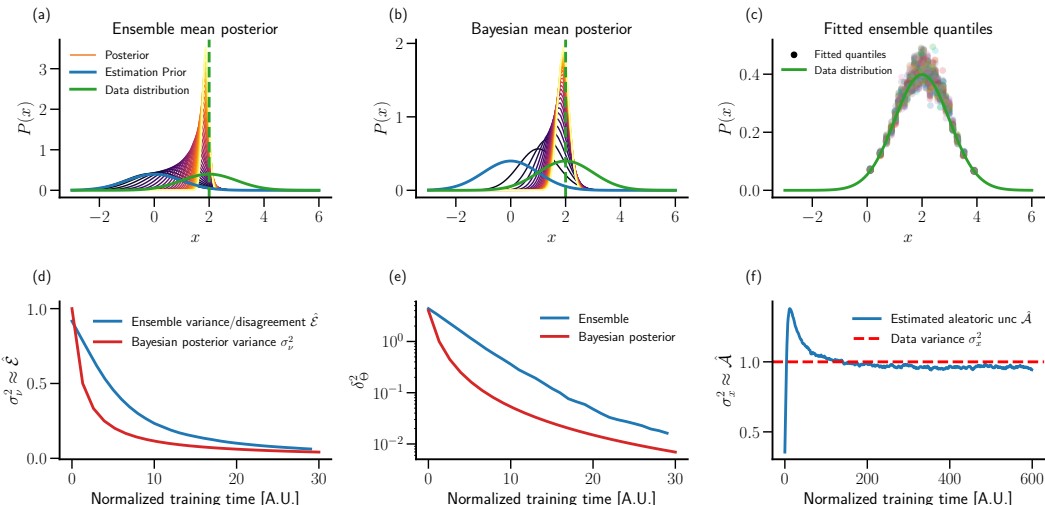

Figure 3: **Variances in the information gain can be approximated by epistemic and aleatoric uncertainty in the information gain**: **(a) and (b)** Evolution during training of the posterior of the mean using an ensemble (gaussian fitted to members of the ensemble at each step) and an ideal Gaussian respectively, as described in Subsection 3.2. Training progresses from purple to yellow. **(c)**: Fitted ensemble quantiles to true data distribution. **(d)**: Ensemble disagreement (equivalent to variance of the posterior estimated with ensemble as $\hat{\mathcal{E}}$ in Equation 8) and true posterior variance $\sigma_\nu^2$ from ideal Gaussian. **(e)**: Distance to the target true value $\delta_\Theta$. **(f)**: Data variance $\sigma_x^2$ approximated with $\mathcal{A}$ in Equation 8. Training time was scaled to show a match between Gaussian posterior and uncertainty measures.

Knowing that the entropy of a Gaussian random variable is $\mathcal{H}(P(\nu)) = 1/2 \log(2\pi e \sigma^2)$, we proceed to compute the information gain (or entropy reduction) of the posterior distribution as

$$\Delta \mathcal{H} = \mathcal{H}(P(\nu)) - \mathcal{H}(P(\nu|x_i)) \tag{37}$$

$$= \frac{1}{2} \log\left(2\pi e \sigma^2\right) - \frac{1}{2} \log\left(2\pi e \left(\frac{\sigma^2 \sigma_x^2}{\sigma_x^2 + \sigma^2}\right)\right) \tag{38}$$

$$= \frac{1}{2} \log\left(1 + \frac{\sigma^2}{\sigma_x^2}\right). \tag{39}$$

We consider $\sigma^2 = \hat{\mathcal{E}}_\delta$ as a form of epistemic uncertainty that can be reduced by sampling more points, and $\sigma_x^2 = \hat{\mathcal{A}}$ as aleatoric uncertainty, which is the underlying irreducible noise of the data, giving a prioritisation variable

$$p_i = \Delta \mathcal{H}_\delta = \frac{1}{2} \log\left(1 + \frac{\hat{\mathcal{E}}_\delta(s,a)}{\hat{\mathcal{A}}(s,a)}\right). \tag{40}$$

As discussed in the main text, other form of priority variables $p_i$ can be effective in some settings. We extend the discussion about uncertainty ratios in the following sections, and show empirical results in the arm bandit task in Appendix E.

### D.2 VARIANCE AS UNCERTAINTY ESTIMATION

To justify our choice of $\sigma^2 = \hat{\mathcal{E}}$ and $\sigma_x^2 = \hat{\mathcal{E}}$ in the information gain described in Equation 11, we train an ensemble of distribution regressors to learn the mean from Gaussian samples ($\mu_x = 2$, $\sigma_x = 1$). This ensemble is compared to the Bayesian posterior distribution of the mean (Gaussian prior, likelihood, and posterior) as detailed in Subsection 3.2. The ensemble, composed of 50 distribution quantile regressors, is initialized with the same prior as the Bayesian model – a unit variance Gaussian centered at 0 – by sampling 50 values from this prior and setting the initial mean of each quantile regressor accordingly. Both the ensemble and Bayesian models are trained using

samples from the data distribution. The ensemble training process follows the method described in the paper, and where each regressor is updated with a probability of 0.5 to introduce ensemble variability. The updates are performed using quantile regression as outlined in Subsubsection 2.3.2. At each time step, the ensemble's estimated posterior is computed by averaging the means of all regressors and calculating the variance of these means.

Figure 3 (a) and (b) illustrate the posterior evolution of both models from the same starting prior, given more samples. Both posteriors exhibit similar trends (the Bayesian model converges faster to the mean, due to the use of TD-updates with a smaller learning rate in the ensemble). In the Bayesian model, posterior sharpness is quantified by its variance, $\sigma_\nu^2$, whereas for the ensemble, it corresponds to the epistemic uncertainty $\hat{\mathcal{E}}$ from Equation 8. Both measures converge to zero, but at different rates Figure 3d. The aleatoric uncertainty of the data, by definition the variance $\sigma_x^2$, is well approximated by $\hat{\mathcal{A}}$ from Equation 8, and shown in Figure 3f. The slight underestimation of the variance is a known issue in quantile regression, as quantiles often fail to capture lower probability regions (Figure 3c), leading to an underestimation of the distribution's variance. Our contribution to prioritization involves incorporating the distance to the target $\delta_\Theta$ from Equation 9 (Figure 3e). This approach prioritizes transitions not only based on the reduction in posterior variance but also on the regressor's proximity to the target.

### D.3 UNCERTAINTY RATIOS

Having arrived at various methods for estimating epistemic and aleatoric uncertainty using distributional reinforcement learning, we now consider how to construct prioritisation variables from these estimates. Naively, one might consider prioritising directly using the epistemic uncertainty estimate; but neglecting the inherent noise or aleatoric uncertainty entirely ignores the 'learnability' of the data. Many methods in related learning domains can be interpreted as incorporating both uncertainties, including Kalman learning Welch et al. (1995); Gershman (2017), active learning Cohn et al. (1996), weighted least-squares regression Greene (2000), and corresponding extensions in deep learning and reinforcement learning Mai et al. (2022). To gain an intuition on how the choice of functional form might impact our particular use-case of prioritisation for various magnitudes of epistemic and aleatoric uncertainty.

$\mathcal{E}/\mathcal{A}$ has desirable properties. For instance under Bayesian learning of Gaussian distributions, $\log(1 + \mathcal{E}/A)$ maximises information gain (see Subsection 3.2), but discontinuities around very low noise must be dealt with—for instance by adding small constants to the denominator. Normalising instead with the total uncertainty is another way of handling the discontinuities. $\mathcal{E}^2/\mathcal{U}$ in particular corresponds to maximising reduction in variance under Bayesian learning in the same Gaussian setting. Both of these forms have the advantage over e.g. $\mathcal{E}/\mathcal{A}$ of preferring low epistemic uncertainty for equal ratios of epistemic and aleatoric uncertainties, i.e. they are not constant along the diagonal of the phase diagram. More generally, it is difficult to say *a priori* which functional form is optimal. Many factors, including the data distributions, model and learning rule will play a role. Further discussion on these considerations can be found in Subsection D.4. These trade-offs are also borne out empirically in the experimental Section 4 & Section 5 below.

### D.4 BIAS AS TEMPERATURE

Lahlou et al. (2022) and others make an equivalence between excess risk and epistemic uncertainty. Concretely, if $f^*(x)$ is the Bayes optimal predictor, the excess risk is defined as:

$$\text{ER}(f, x) = R(f, x) - R(f^*, x), \tag{41}$$

where $R$ is the risk and $R(f^*, x)$ can be thought of as the aleatoric uncertainty.

One possible issue arises in overstating the connection between excess risk and epistemic uncertainty. Consider the case where there is model mis-specification, and $f^*$ is not in the model class; then assuming the model class is fixed (as is standard), then the lower bound of $\text{ER}(f, x)$ is non-zero. Stated differently, it is *not* fully reducible, which is often viewed as a central property of epistemic uncertainty. For some applications this distinction may not be important; there is some non-zero lower bound to the epistemic uncertainty but the ordering and correlations are intact under this equivalence. But it could also play a significant role. For us in particular, adopting this equivalence has two related consequences:

1. The model mis-specification acts as a temperature for our prioritisation distribution;

2. The ratio, or more generally the functional form of our prioritisation variable, can offset this temperature.

To make the above equation fully reducible, we would need to further subtract a term capturing the difference between the Bayes predictor, and the best predictor in the model class i.e. the model bias or mis-specification term. Let us denote this term by $C$, and assume it constant over the domain. And let us denote the fully reducible uncertainty by $\eta$. In the case where we use the excess risk, the prioritisation of sample $i$ is given by

$$[\text{Vanilla}] \qquad p_i = \frac{\eta_i + C}{\sum_i (\eta_i + C)} = \frac{\eta_i + C}{NC + \sum_i \eta_i}. \tag{42}$$

It is easy to see how $C$ acts as a temperature. In the limit of large $C$ we get a uniform distribution over samples. Similarly if $C = 0$ we recover the 'true' distribution for reducible uncertainty.

It is of course hard to measure this model mis-specification term. In large networks we can assume the capacity is unlikely to be restrictive, but perhaps other parts of the training regime could play a part. Importantly, the above holds true not just for model mis-specification, but also if there is any systematic error in the epistemic uncertainty estimate (i.e. think of C as an error on the epistemic uncertainty estimate).

### D.5 PRIORITISATION DISTRIBUTION ENTROPY

Assuming the above effect is significant, might a different functional form (as discussed in Subsection D.3) for prioritisation alleviate the impact? Consider the following additional options:

$$[\mathcal{E}/\mathcal{U}] \qquad p_i = \frac{\frac{\eta_i + C}{\eta_i + C + \beta_i}}{\sum_i \frac{\eta_i + C}{\eta_i + C + \beta_i}}; \tag{43}$$

$$[\mathcal{E}^2/\mathcal{U}] \qquad p_i = \frac{\frac{(\eta_i + C)^2}{\eta_i + C + \beta_i}}{\sum_i \frac{(\eta_i + C)^2}{\eta_i + C + \beta_i}}; \tag{44}$$

and more generally,

$$[\mathcal{E}^m/\mathcal{U}] \qquad p_i = \frac{\frac{(\eta_i + C)^m}{\eta_i + C + \beta_i}}{\sum_i \frac{(\eta_i + C)^m}{\eta_i + C + \beta_i}}. \tag{45}$$

In the limit of large $C$ all of these forms tend to a uniform distribution. However, at what rate? And is there anything else interesting we can say?

Consider the following toy problem:

- Populate "replay" buffer with $N$ samples;
- Each sample's reducible uncertainty is sampled from $\rho_\eta$;
- Each sample's reducible uncertainty is sampled from $\rho_\beta$;
- $C$ is constant over the samples.

We can plot as a function of $C$ the entropy of the prioritisation distribution for the functional forms above. Such a plot is shown for various choices of $\rho_\eta$, $\rho_\beta$ in Figure 4. Clearly, as $C$ increases the entropy in the distribution increases and saturates at some maximum entropy. There is some variation in the entropy ordering depending on the exact $\rho_\eta$, $\rho_\beta$ distributions; in some instances the vanilla form is lower entropy than $\mathcal{E}/\mathcal{U}$, but in general the entropy remains lower for longer (as a function of $C$) when the exponent in the nominator is higher. This is not a particularly surprising result, but lends support to the idea that a higher order function of $\mathcal{E}$ in a ratio form is desirable for prioritisation.

### D.6 RELATION TO $\mathcal{E}$ UNDER 0 BIAS

Now let us consider a more interesting measure. Ordinarily, or naively—in the sense that this is the first order approach—we want our prioritisation variable to be the vanilla prescription; and ideally

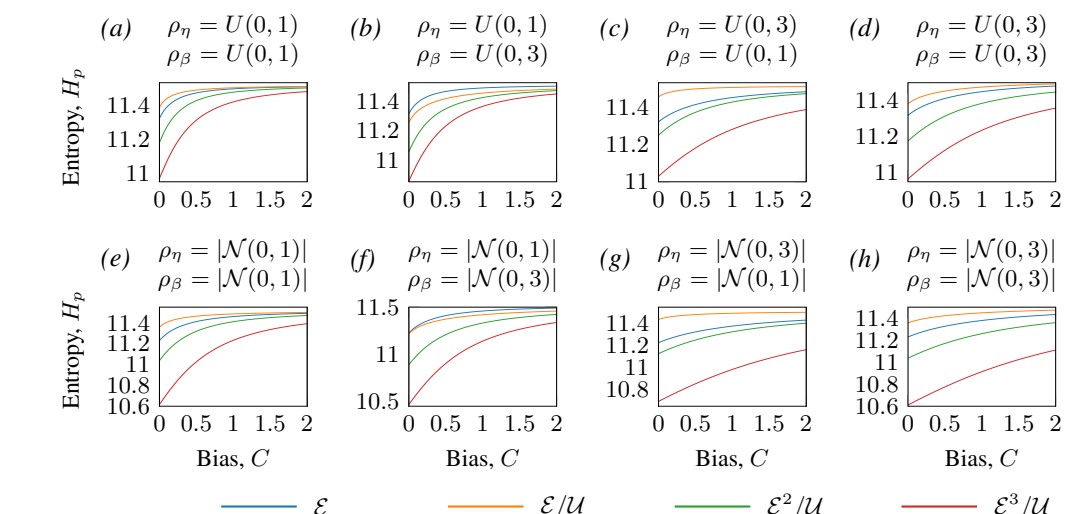

Figure 4: **Ratios can reduce entropy of distribution under bias.**

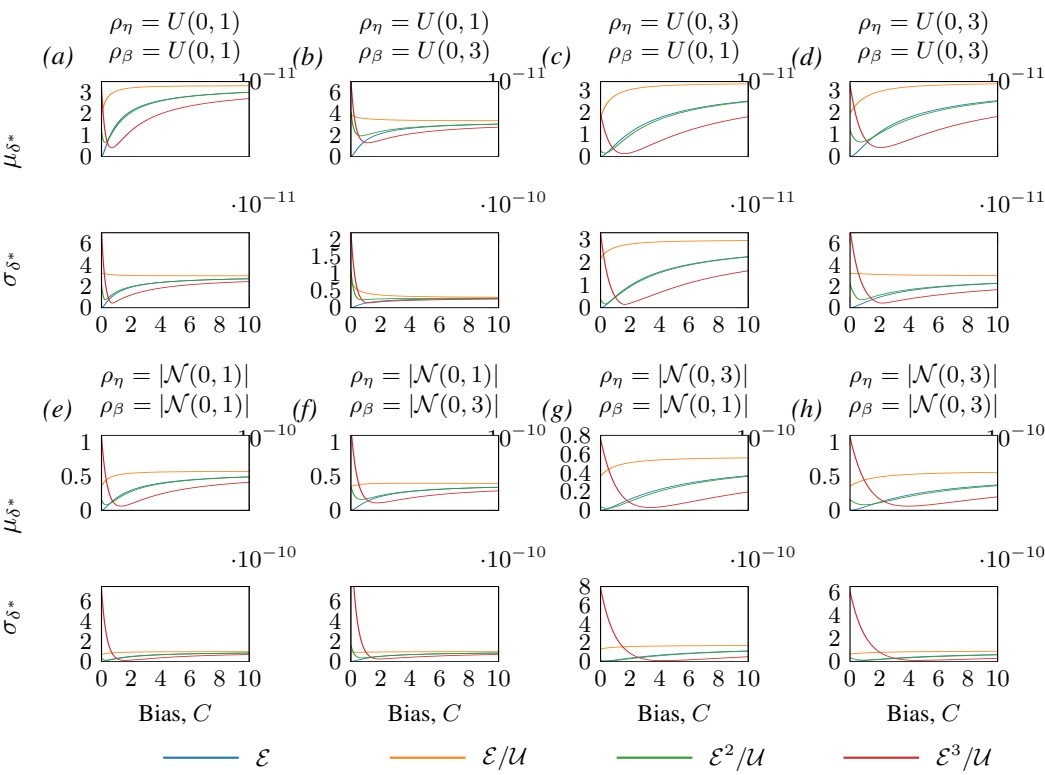

Figure 5: $\mathcal{E}^2/\mathcal{U}$ **closely approximates** $E$ **for non-trivial bias.**

we would want $C$ to be 0. We can measure the difference, which we denote $\delta_i$ to this ideal for each functional form as a function of $C$. This plot is show for various choices of $\rho_\eta$, $\rho_\beta$ in Figure 5.

In general, the standard $\mathcal{E}/\mathcal{U}$ ratio is poor, it has systematically higher mean and variance of error. Beyond that, a clear trade-off emerges: as you increase the exponent $m$, then for high $C$ there is lower deviation from the 'correct' distribution for priority. This is related to maintaining lower entropy and tending to a uniform distribution more slowly. However, for lower $C$ you are likely to be more wrong, catastrophically so. This trade-off for $m = 3$ is effectively crossed when the red line intersects with

the blue in these plots. The point at which this intersection happens will be a function of various things, primarily the underlying distributions—in this case $\rho_\eta$, $\rho_\beta$.

Interestingly however, for $m = 2$ there is very fast convergence of $\mathcal{E}^2/\mathcal{U}$ and $\mathcal{E}$ as a function of $C$. So while $m = 3$ has a very stark trade-off, $m = 2$ is less extreme: For low $C$ it may make you more wrong but generally you will have very similar average error by this metric to the vanilla case; all the while the entropy of the distribution will be much lower and more informative (as shown in Figure 4). This toy model is clearly very simplistic, not least the lack of variation in $C$ over the samples; but future work could be dedicated to understanding these trade-offs more formally in the context of prioritized replay.

### D.7 OFF-SETTING BIAS WITH TD TERM

Leaving aside the ratio forms, the consequences of the temperature effect may differ depending on the choice of epistemic uncertainty estimate we use. The methods we discuss in Section 2 & Section 3 all effectively use the equivalence of excess risk and epistemic uncertainty, and so do not explicitly consider the possibility of model bias. The possible exception is the method resulting from the expansion of the average error over the quantiles and ensemble in Subsection 3.1. The main difference between this decomposition and that of Clements et al. (2020) is a term that encodes the distance from the target:

$$\delta_\Theta^2 = \left( \Theta - \mathbb{E}_{\psi, i} \left[ \theta_i(\psi) \right] \right)^2 . \tag{46}$$

This term *could* guard against two possible shortcomings of the decomposition in Clements et al. (2020):

1. Consider the pathological case in which each ensemble is initialised identically, then each quantile will have zero variance and the epistemic uncertainty measure from Clements et al. (2020) will be zero. Even if there is independence at initialisation, there may be characteristic learning trajectories or other systematic biases that push the ensemble together and lead to an underestimate in epistemic uncertainty. Here, the term above—if treated as part of the epistemic uncertainty—can continue to drive learning in ways we want.

2. However, it could be that the ensemble behaves nicely and the metric over the ensemble from Clements et al. (2020) is principally a good one, *but* that there is significant model bias. This could also be captured by the term above but would need to be *subtracted* from the total error in order to get a fully reducible measure for epistemic uncertainty (as per the argument discussed above).

Which of the two problems is more pronounced is difficult to know *a priori*, and could be an avenue for future work. Empirically, the performance of the UPER agent in Section 5 suggests that the former is the greater effect—at least on the atari benchmark with the model architecture and learning setting used.

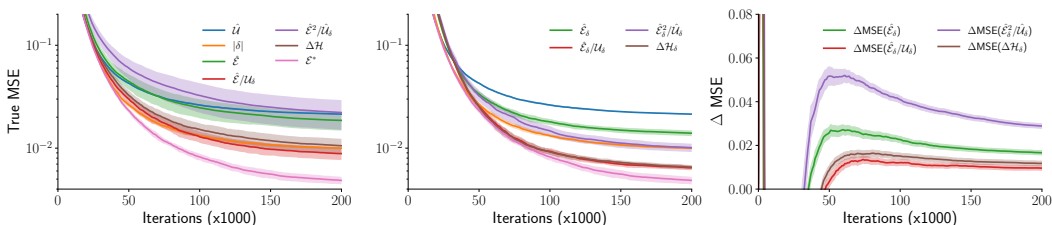

Figure 6: Comparison of MSE for different prioritisation scheemes. Left panel, shows ratios and information gain based on epistemic uncertainty $\hat{\mathcal{E}}$ proposed by Clements et al. (2020). Middle panel, shows ratios and information gain based on our proposed *target epistemic uncertianty* $\hat{\mathcal{E}}_\delta$. Right panel, different in MSE between curves in the left panel and right panel for the shifted arm task. For instance, $\Delta\text{MSE}(\hat{\mathcal{E}}_\delta) = \text{MSE}(\hat{\mathcal{E}}) - \text{MSE}(\hat{\mathcal{E}}_\delta)$, showing that our proposed $\hat{\mathcal{E}}_\delta$ is in general better for prioritisation in the arm-bandit task. Averaged across 10 seeds.

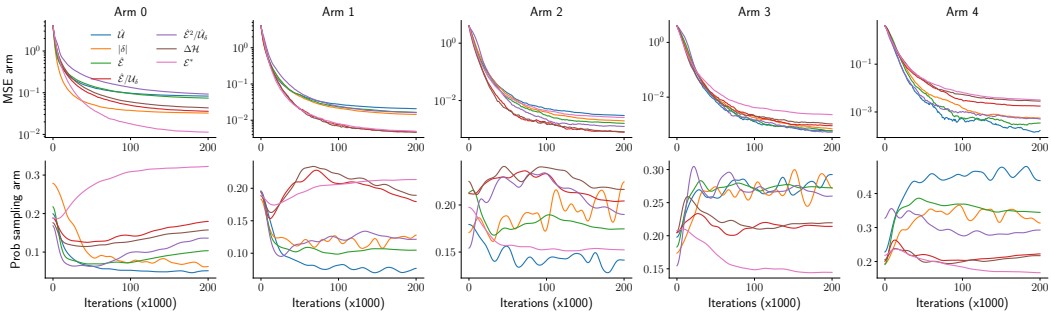

Figure 7: Comparison of MSE for different prioritisation scheemes using $\hat{\mathcal{E}}$ based prioritisation. Total uncertainty $\mathcal{U}$ and TD-error prioritisation tend to oversample high variance arms compared to epistemic uncertainty prioritisation.

# E ARM-BANDIT TASK

The hyperparameters used in the Arm-Bandit Task shown in section 4 are shown below:

- Number of train steps: $20^5$
- Learning rate annealing: $0.005 \cdot 2^{-\text{iters}/40000}$.
- Init variance estimation: uniformly sampled from to 0.1
- Number of agents in the ensemble: 30
- $\alpha = 0.7$, $\beta$ is annealing from 0.5 to 1 in 0.4 to 1 in proportional prioritisation as in the original work by Schaul et al. (2016).
- n arms: $n_a = 5$, $\bar{r} = 2$, $\sigma_{\max} = 2$ and $\sigma_{\min} = 0.1$.
- Number of quantiles: 30.
- Quantiles initialized as uniform distribution between -1 and 1. For the main results in **??**, $\theta_\tau$ are initialized randomly between -1 and 1, then sorted to describe a cumulative distribution.
- Each agent in the ensemble is updated with probability 1/2 on each step.
- For the shifted arm experiment, the mean reward per arm $\bar{r}(a) = 3, 2.75, 2.5, 2.25, 2$ for arms $1, 2, 3, 4$ and $5$.

Figure 6 show the mean squared error from the estimated $Q(a) = \mathbb{E}_{j,\psi}\left[\theta_j(\psi)\right]$ to the true mean, where $\psi$ denotes agents in the ensemble case. Figure 8 and Figure 8 show the probability of sampling each arm from the memory buffer throughout the training, and the mean square error from the estimated arm value $Q(a)$ to the true arm value $\bar{r}$ (the same for every arm). In addition, we depict the evolution of uncertainty quantities for all prioritisation variables for the arm bandit task in Figure 9.

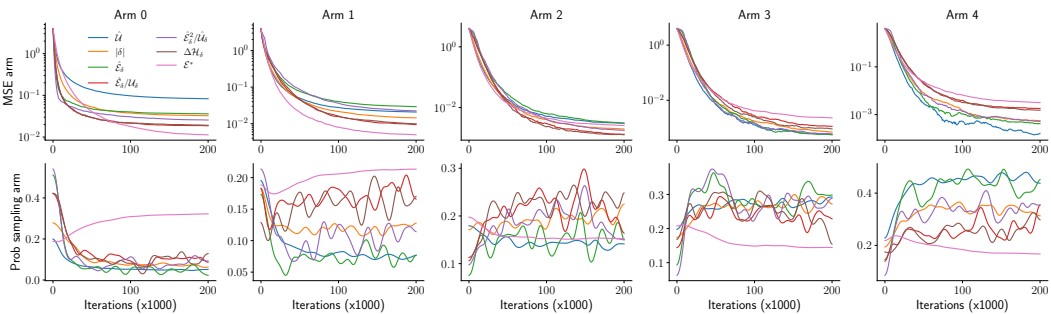

Figure 8: Comparison of MSE for different prioritisation scheemes using $\hat{\mathcal{E}}_\delta$ based prioritisation. Total uncertainty $\mathcal{U}$ and TD-error prioritisation tend to oversample high variance arms compared to epistemic uncertainty prioritisation.

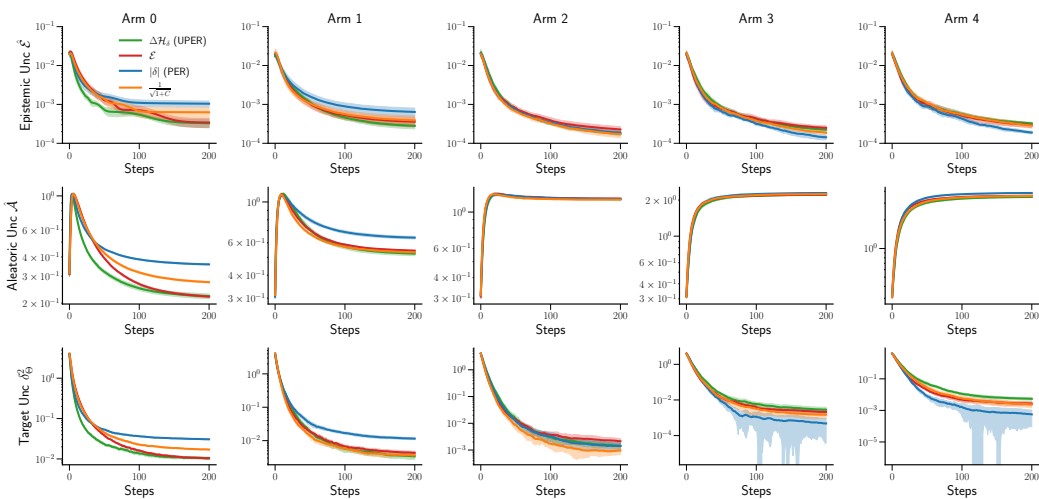

Figure 9: Epistemic uncertainty $\hat{\mathcal{E}}$ and target uncertainty $\delta_\Theta^2$ decrease more rapidly for lower noise arm (first column), for UPER compared to other methods. The inclusion of aleatoric uncertainty in the prioritization variable, as utilized in the information gain formula, aims to sample transitions with high epistemic uncertainty for its reduction, while also avoiding transitions with high aleatoric uncertainty with less learnable content. This rationale is reflected in the ratio presented in the derived $\Delta\mathcal{H}_\delta$, and shown its effect in the sampling probabilities plotted in Figure 8. The TD-error tends to oversample noisier transitions, resulting in less frequent updates for the least noisy arm, consequently leading to higher levels of epistemic and target uncertainty for that arm.

## F  GRIDWORLD EXPERIMENTS

The hyperparameters used in Figure 1 are listed below:

- Learning rate: 0.1
- Discount factor, $\gamma$: 0.9
- Exploration co-efficient, $\epsilon$: 0.95
- Buffer capacity: 10,000
- Episode timeout: 1000 steps
- Random reward distribution: $\mathcal{N}(0, 2)$

For every 10 steps of 'direct' interaction and learning from the environment, the agent makes 5 updates with 'indirect' learning from the buffer replay. The data shown in the plots consists of 100 repeats and is smoothed over a window of 10.

# G  ATARI EXPERIMENTS

Cumulated training improvement of UPER over PER, QRDQN, QR-PER and QR-ENS-PER are shown in Figure 11 to Figure 14. The accumulate percent improvement $C_{\text{UPER/PER}}$, (same for $C_{\text{UPER/QRDQN}}$ and the rest), is computed as

$$C_{\text{UPER/PER}} = \frac{\sum_t \left[ \text{UPER}_{\text{human}}(t) - \text{PER}_{\text{human}}(t) \right]}{\sum_t \text{PER}_{\text{human}}(t)} \cdot 100 \tag{47}$$

where $t$ indexes training time, and $\text{UPER}_{\text{human}}$ (same for $\text{PER}_{\text{human}}$ and $\text{QRDQN}_{\text{human}}$) denotes human normalized performance.

For the baseline experiments we use the same implementations as those of the original papers, including hyperparemeter specifications. For our UPER method, we performed a limited hyperparameter sweep over 3 key hyperparameters: learning rate and $\epsilon$ for the optimizer, and the priority exponent. The sweep ranged $3 \times 10^{-5}$ to $5 \times 10^{-5}$ for the learning rate, $6.1 \times 10^{-7}$ to $3.125 \times 10^{-4}$ for $\epsilon$ and 0.6 to 1 for the priority exponent. We chose values for our final experiments based on average performance over 2 seeds across a sub-selection of 5 Atari games (chopper command, asterix, gopher, space invaders, and battlezone).

## G.1  QR MODELS ABLATION

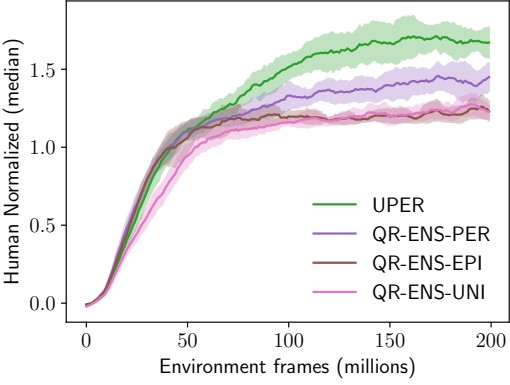

Figure 10: Comparison of ablated prioritization variables. Median Human Normalized Score for QR-DQN ensembles, where only the prioritization variable is changed. UPER, PER, EPI, and UNI use the information gain in Equation 11, the TD-error, target epistemic uncertainty in Equation 13, and uniform sampling, respectively.

To demonstrate the effectiveness of the information gain prioritization, and to confirm that the performance improvement stems from our proposed prioritization variable, we compared UPER to identical QR-DQN ensemble agents, maintaining the same architecture but altering only the prioritization variable. The results are presented in Figure 10. UPER outperforms alternative approaches such as QR-DQN-PER, which uses the TD-error to prioritize (as previously shown in Figure 2), QR-ENS-EPI, which directly prioritizes using epistemic uncertainty as defined in Equation 13, and QR-ENS-UNI, which uses uniform sampling. These findings highlight the significance of both epistemic uncertainty and aleatoric uncertainty in prioritizing replay, as included in the information gain term. Additionally, these results confirm that the performance improvement can be solely attributed to the prioritization variable, as the QR-DQN ensemble architecture employed in each agent remains constant.

## G.2  COMPUTATIONAL COST

For the main Atari-57 benchmark results, average clock time training for PER, QR-DQN, and UPER (standard DQN, distributed RL agent, and ensemble of distributed RL agents) are $\approx 150$ hours, $\approx 149$ hours, and $\approx 162$ hours respectively, all implemented in JAX running in Tesla V100 NVIDIA Tensor Cores.

To generate Table 1, we conducted experiments on a laptop equipped with an i5-10500H CPU (2.50GHz) and a 6GB NVIDIA GeForce RTX 3060 Mobile/Max-Q (not the same architecture as the main results in the paper, which uses Tesla V100 NVIDIA Tensor Cores). We ran 40 iterations of Pong for each model, using the last 20 iterations to avoid initialization and buffer filling times. The experiments were conducted on both CPU and GPU using different network architectures. In each iteration, the agent processed 1000 frames and performed one batch update of 64 transitions, with 4 frames per iteration. For all these runs, we used the publicly available implementation of DQN Zoo

by DeepMind. Table 1 shows the time it takes for each iteration (1000 frames and a batch update) in seconds, along with standard deviations. There are two main conclusions from this experiment. First, most of the time consumed during each iteration is spent running the game engine (the 1000 frames per iteration), which is typically run on the CPU. This is evident from the small difference in time between QR-DQN and DQN in both the CPU and GPU cases. This difference could be larger in favor of the GPU if the batch size is increased and the frames per iteration are reduced. Second, we are significantly leveraging the parallelization capabilities of GPUs, as shown by the reduced times for the QR-DQN-ENS model (the architecture needed for UPER) when comparing GPU to CPU performance. The 2-second gap per iteration when comparing QR-DQN-ENS with QR-DQN and DQN is further reduced by utilizing V100 GPUs, as demonstrated by the training times reported in the main Atari-57 experiment.

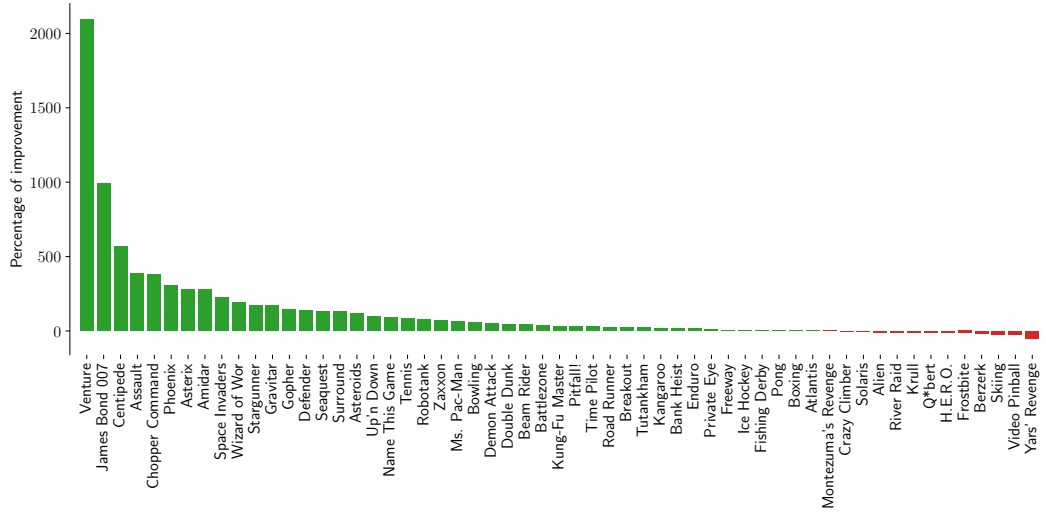

Figure 11: Cumulated training improvement of UPER over PER defined as $C_{\text{UPER/PER}}$.

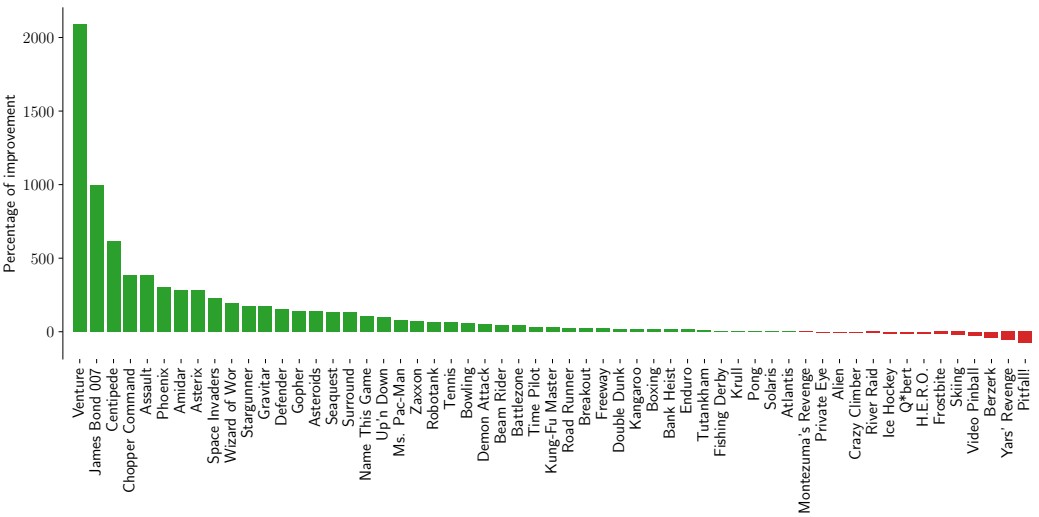

Figure 12: Cumulated training improvement of UPER over QR-DQN defined as $C_{\text{UPER/QR-DQN}}$.

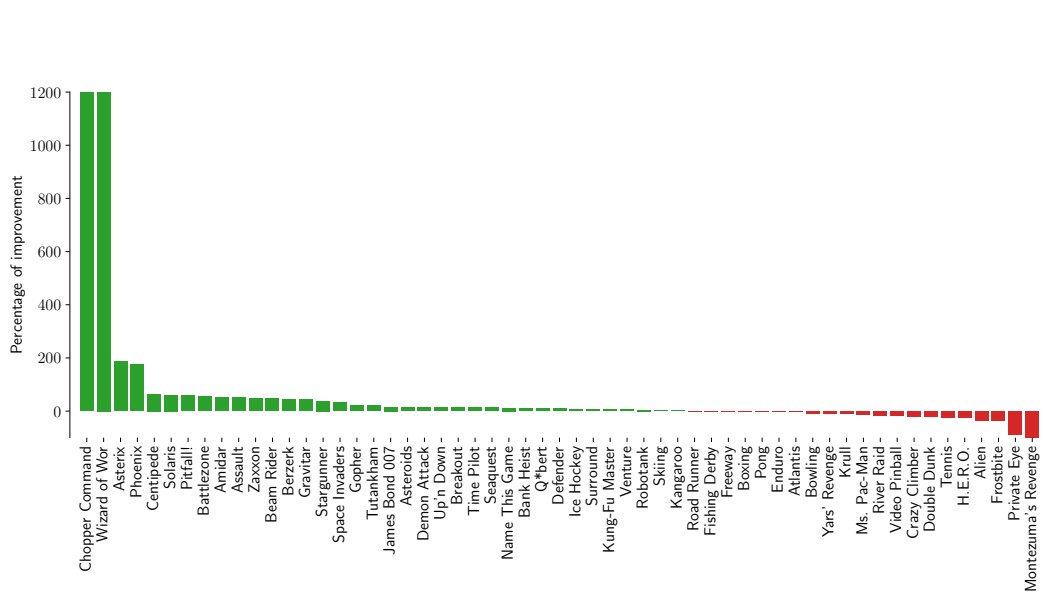

Figure 13: Cumulated training improvement of UPER over QR-PER defined as $C_{\text{UPER/QR-PER}}$.

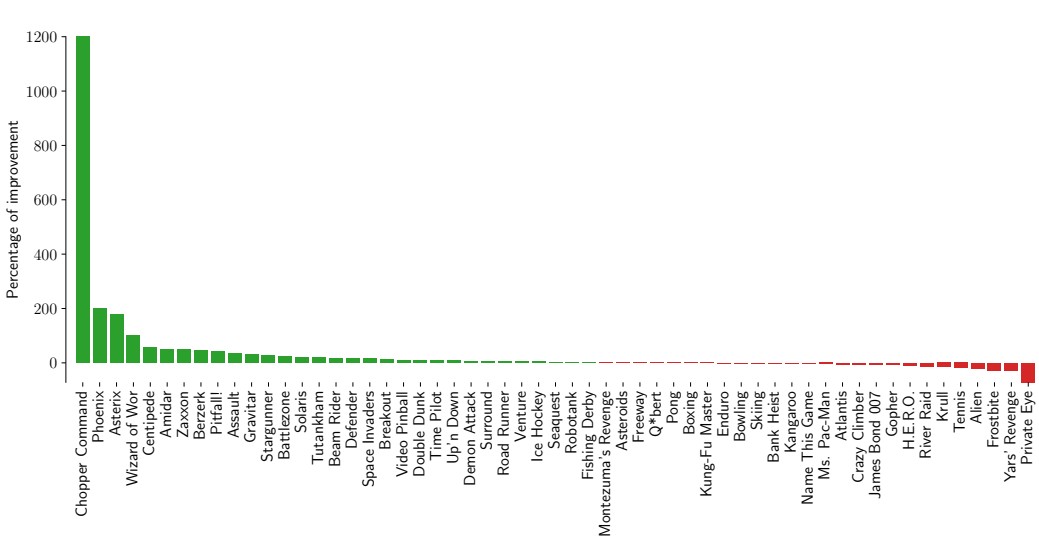

Figure 14: Cumulated training improvement of UPER over QR-ENS-PER defined as $C_{\text{UPER/QR-ENS-PER}}$.

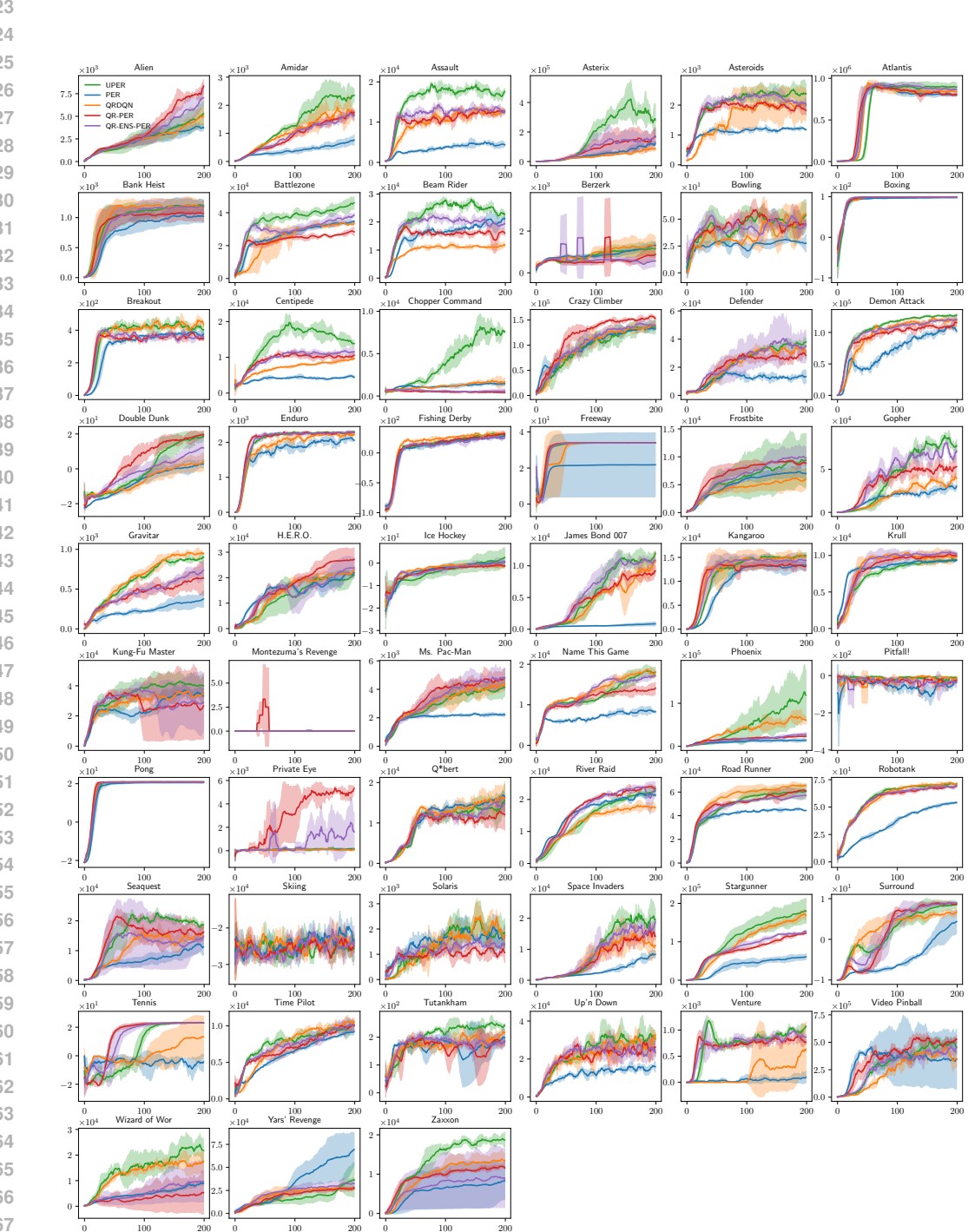

Figure 15: Average performance and corresponding standard deviation for all games across 3 seeds.

