# OpenReview forum: "Uncertainty Prioritized Experience Replay"
_ICLR.cc/2025/Conference — Submitted to ICLR 2025_

### Official Review · Reviewer_pEo7 · 2024-10-24

**Soundness:** 3
**Presentation:** 2
**Contribution:** 2
**Rating:** 5
**Confidence:** 3

**Summary:**

In this paper, the authors propose a new experience replay method, called Uncertainty Prioritized Experience Replay (UPER), which leverages epistemic uncertainty to improve sample efficiency. To estimate epistemic uncertainty, they introduce a modified concept of total uncertainty and they decompose it into aleatoric and epistemic uncertainties, as shown in Eq. (7). Based on the estimation of aleatoric uncertainty, they also propose a priority motivated by information gain, as detailed in Eq. (11). They validated UPER in various atari task under QRDQN

**Strengths:**

1. Reasonable Motivation: The authors introduced two examples: conal bandits and noisy greedy world
2. Easy to Implement but good insights: The authors proposed new formula for uncertainty and a way to compute prioritization using the uncertainty with the concept of information gain.
3. Implementation seems to be not difficult.

**Weaknesses:**

1. The authors used QR-DQN as the benchmark. So it is unclear whether this concept is still valid across different distributional Q-learning (e.g., C51, Rainbow)

2. The authors provided compuational costs in Table 1 for each algorithm. But it is unclear the coupuational cost between Random vs PER vs UPER.

**Questions:**

1. The Introduction section lacks sufficient depth, and the proposed method is introduced too briefly. For example, there is no mention of the need for an ensemble or the importance of the concept of information gain. Additionally, while the motivating examples are significant, they are not presented until Section 4. In my view, the Introduction should consider the following points: First, it would be meaningful to include the motivating examples and explain why epistemic uncertainty is crucial, as well as why PER is not suitable for certain tasks. Second, providing why the concept of information gain is necessary would improve the readibility. Third, utilizing an ensemble, while the computational costs are marginal, to estimate the uncertainties would be also helpful to the readers. I would appreciate it if the authors could provide their thoughts on these points.


2. It could be valuable to evaluate the feasibility of this method under different RL distributions. For example, the study (https://arxiv.org/abs/1906.05243) referenced in this paper used only 100K steps (see Table E). I would appreciate your thoughts on whether the experiments I have proposed are feasible by utilizing the efficient rainbow in (https://arxiv.org/abs/1906.05243) or C51 another algorithm, even though  the new method seems to be effectively independently of algorithms from a mathematical standpoint.

3. Prioritized Experience Replay (PER) is a meaningful method, but I believe that some of the expressions used may contain exaggerated aspects. For instance, the authors state, "PER has been widely adopted as a standard technique in DRL" in Line 29, and "Prioritized experience replay is a crucial component of contemporary deep reinforcement learning models" in Line 10. However, I think that the effectiveness of PER has been primarily studied in value-based learning, rather than in policy-based or actor-critic methods, such as TD3. I would appreciate hearing the authors' perspective on this matter.

4. I believe the readability of this paper could be improved if the authors were to change their citation style, such as using numbering or placing parentheses around citations. For instance, in line 39~41, "PER is an extension of Experience Replay Lin (1992), which uses ...." can be revised as "PER is an extension of Experience Replay, as introduced by Lin (1992)" or "PER is an extension of Experience Replay (Lin (1992))". I am curious about the authors' opinions.

**Details Of Ethics Concerns:**

No Ethics Concerns

---

> ### Author Response · Authors · 2024-11-19
>
> We thank the reviewer for their constructive review. We are glad you find the method insightful. We hope to address some of your concerns around clarity, phrasing, and evaluations below.
>
> > W1 [also addresses Q2]: Our prioritization scheme is not limited to use with quantile representations of the distribution. As noted in your review the method is in principle independent of the underlying algorithm. However, the quality of our epistemic and aleatoric uncertainty estimates do matter. The specific algorithms you mention (C51 and rainbow) are categorical distributional algorithms, which are well known to overestimate the variance when used in discounted bootstrapping (see for instance Fig. 2 of Rowland et. al 2019). This overestimate will significantly impact the quality of our information gain criterion and so we do not expect UPER to perform well with C51 or rainbow. We would fully expect a quantile regression version of rainbow to work well with UPER, but developing this agent was beyond the scope of this paper.
>
> > W2: We appreciate the possible confusion associated with Table 1. It is worth clarifying that the computational costs associated with the methods we are discussing can be decomposed into two parts: first is the cost associated with performing prioritized sampling. These costs are discussed in the original PER paper, and amount to approximately “2-4% increase in running time and negligible additional memory usage”. This part of the cost is identical in our method, so we do not include our own analysis. The second component of the cost pertains to the computation of the prioritization variables themselves. In PER there is no additional cost in this component compared to vanilla DQN since the TD error is already computed. In our method however we require an ensemble of distributional estimators to compute our information gain prioritization criterion. This cost is the focus of the comparisons shown in Table 1. Since the majority of our network is shared across distribution and ensemble heads, the additional cost is insignificant, particularly when leveraging parallelization of GPU operations. We have added a sentence to the manuscript to clarify the implications of Table 1.
>
> > Q1: We appreciate the feedback on the readability of the introduction. We have updated the introduction to be more comprehensive and better sign-post our contributions. In particular, we give intuitions for the use of epistemic and aleatoric uncertainty under the information gain criterion in prioritising replay. In addition to the noisy TV thought experiment borrowed from the exploration literature to explain the potential shortcomings of PER, we discuss a toy game involving estimating two distributions to help motivate the intuition behind the information gain. We also mention more explicitly and sooner in the narrative the particulars of our methodology i.e. ensembles and distributions. Hopefully, this helps to address your concerns regarding the depth of the introduction.
>
> > Q3: This is a fair distinction to raise. We have amended the phrasing to ‘Prioritized experience replay is a crucial component of contemporary value-based deep reinforcement learning models’. On the broader point about PER in policy-gradient methods: it has generally been difficult to use PER in policy gradient methods, especially when training the actor (Saglam et. al 2024). However approaches such as phasic policy gradient methods (Cobbe et. al 2020), which splits training into two phases, may allow prioritizing when training the critic without high variance updates from large error transitions harming training of the policy network.
>
> > Q4: Thank you for catching these oversights in the style of some of our citations. We have gone through the manuscript and updated the citation commands to match the ICLR guidelines (as outlined here: https://arxiv.org/html/2409.06957v1). We hope this improves the readability of the paper–for instance in the introductory passage you highlighted.
>
> __Refs__
>
> Statistics and Samples in Distributional Reinforcement Learning (Rowland et. al 2019)
>
> Actor Prioritized Experience Replay (Saglam et. al 2024)
>
> Phasic Policy Gradient (Cobbe et. al 2020)

---

> ### Comment · Reviewer_pEo7 · 2024-11-21
>
> Thank you for providing meticulous revisions and responses to my requests.
>
> However, it is still crucial to demonstrate that the proposed method is effective across various RL distributional methods, even if some results show only marginal improvement. Otherwise, the method will be viewed as being overly specialized for QR-DQN, thus, its impact may be regarded as too limited by the research community.

---

> ### Author Response · Authors · 2024-12-04
>
> Thanks for your response. We are glad to have been able to address the majority of your concerns. The extended rebuttal period did give us the necessary time to perform some additional experiments. Please see our general response that shows significant improvements to C51 with UPER. Thanks again for your engagement and feedback. We hope these experiments address the last of your concerns.

---

### Official Review · Reviewer_7m4h · 2024-11-02

**Soundness:** 3
**Presentation:** 2
**Contribution:** 2
**Rating:** 5
**Confidence:** 3

**Summary:**

This paper proposes a novel prioritized experience replay method to enhance exploration in reinforcement learning (RL). To address the adverse effects of noise on value estimation observed in previous methods, the authors suggest leveraging epistemic uncertainty to guide the prioritization of transitions. The effectiveness of their approach is demonstrated through comprehensive experiments.

**Strengths:**

1. The paper tackles the critical issue of exploration in RL and offers a promising method that uses uncertainty to prioritize interaction data.
2. Comprehensive experiments are provided, ranging from bandits and tabular tasks to Atari tasks.

**Weaknesses:**

1. The paper lacks contemporary baselines in Atari games. The baselines compared in this paper are somewhat outdated, while several recent works also consider estimating uncertainty to improve exploration.
2. The writing could be polished further. Some sentences are quite colloquial, such as "this is shown graphically in Figure 14 and Figure 15" in line 394.
3. The paper primarily discusses the advantages of their prioritized variables through verbal descriptions and toy examples. It would benefit from some theoretical analysis to substantiate the improvement of their methods.

**Questions:**

1. I don't think the MSE in Figure 1(b) is a good metric for bandit tasks, as overfitting can also lead to lower MSE. Can you replace it with accumulated regret? Figure 1(c) is messy. The legend in the figures is "Unstable" and "Stable," but the caption uses "stablest" and "noisiest." The colors in the legend are black, but the lines in the figures are blue and green. Figure 1(c) needs to be polished with clearer explanations.
2. More baselines should be considered in Atari. The naive prioritized DQN[1] can act as the basic baseline since it uses the TD-error as the priority. Additionally, other methods using uncertainty should be considered, such as HyperAgent[2]. There are some Atari games requiring hard exploration as described in [3]. How does your method perform on these Atari games? Can UPER be applied to some SOTA deep RL algorithms, such as Rainbow[4] or BBF[5]?
3. It seems like the uncertainty in Eq(9) also depends on the error. Does it consider the disagreement among ensembles?

[1] Schaul T. Prioritized Experience Replay[J]. arXiv preprint arXiv:1511.05952, 2015.

[2] Li Y, Xu J, Han L, et al. Q-Star Meets Scalable Posterior Sampling: Bridging Theory and Practice via HyperAgent[J]. arXiv preprint arXiv:2402.10228, 2024.

[3] Bellemare M, Srinivasan S, Ostrovski G, et al. Unifying count-based exploration and intrinsic motivation[J]. Advances in neural information processing systems, 2016, 29.

[4] Hessel M, Modayil J, Van Hasselt H, et al. Rainbow: Combining improvements in deep reinforcement learning[C]//Proceedings of the AAAI conference on artificial intelligence. 2018, 32(1).

[5] Schwarzer M, Ceron J S O, Courville A, et al. Bigger, better, faster: Human-level atari with human-level efficiency[C]//International Conference on Machine Learning. PMLR, 2023: 30365-30380.

---

> ### Author Response · Authors · 2024-11-19
>
> Thank you for your detailed review and questions. We appreciate that you found our work novel and supported by comprehensive experiments. However, we realize we could have been clearer about the goal of our proposed method.
>
> UPER modifies the prioritization variable to account for uncertainty when sampling transitions from the memory buffer for updating the agent’s Q-value estimates. It does not explicitly enhance exploration or alter the agent's interaction with its environment, though prioritizing uncertainty-based transitions might have such effects indirectly. As a result, while UPER is related to exploration literature, we do not consider it an exploration algorithm, as we do not promote exploration directly through methods like rewarding unvisited, uncertain, or high-error states. Instead, our focus is on how transitions are sampled from the buffer for action-value function updates, borrowing ideas from the exploration-exploitation trade-off literature to estimate the quantities needed for our method, epistemic and aleatoric uncertainty. We extended the introduction and discussion section to clarify this. We provide detailed answers to your comments below and hope this clarification helps highlight the contribution of our method. Please let us know any further concerns.
>
> > Comparison with other baselines
>
> In [3], the authors use pseudo-counts to quantify state uncertainty. This measure does not distinguish between epistemic and aleatoric uncertainty but is instead used to reward unvisited states with an exploration bonus (as described in the first equation of [3]). Unlike this approach, UPER does not modify state rewards based on novelty or uncertainty. Regarding the application of our method to C51, BBF, or Rainbow, methods built on categorical distribution estimation, we do not expect UPER to perform well. This is because these agents rely on fixed support for categorical distribution estimation, and when bootstrapping with discounting, the target distribution needs to be projected onto the support of the estimated distribution; here the variance is inherited and tends to be overestimated. This is a well documented phenomenon (see e.g. Rowland et. al 2019). It may have been possible to try UPER on a quantile regression version of Rainbow, this would have required developing a new agent entirely, which we felt was beyond the scope of this paper. Meanwhile, QR-DQN is generally a far stronger algorithm than C51 (see original QR-DQN paper).
>
> > Q1: Thank you for noticing this. The MSE shown in Figure 1b is the average across all arms, calculated between the estimated mean and the true mean (i.e., the mean used to sample rewards from the arm). By definition, this represents the expected generalization error and is not subject to overfitting, and it was used to show the performance for each method exactly for this reason. We included this explanation in the text. Additionally, we revised Figure 1 and its explanations to improve clarity, as well as the other suggested changes in the writing style.
>
> > Q2: We compare our method with the one you refer to as “naive prioritized DQN” [1]. In our work, we call it Prioritized Experience Replay (PER), as shown in Figure 2 and mentioned in the first paragraph of Section 5. Regarding Atari games that require extensive exploration, as we stated earlier, our method does not explicitly promote exploration. It only alters the prioritization variable to sample transitions from the replay buffer. Therefore, we do not expect our method to improve performance on these games, as it was not designed to incentivize exploration explicitly. One example of such a game is Montezuma’s Revenge, as mentioned by reviewer C2gj. This game has been solved by explicitly incentivizing exploration, such as by rewarding unvisited, high-uncertainty, or high-error states, or through imitation learning. For a review of these methods and ideas, please refer to our response to reviewer C2gj, W2.
>
> > Q3: Yes, you are correct. The total uncertainty $\hat{\mathcal{U}}{\delta}$ in Equation 9 depends on the error $\delta{\Theta}$, the ensemble disagreement (epistemic uncertainty) $\hat{\mathcal{E}}$, and the aleatoric uncertainty $\hat{A}$. Here, $\delta_{\Theta}$ represents the TD error between the true target and the combined estimation from all quantiles and ensembles. Considering the expected error across quantiles and ensembles as total uncertainty introduces an extra term, yielding the target epistemic uncertainty $\mathcal{E}{\delta}$, which accounts for both the ensemble disagreement and the distance to the estimation quantity. Compared to previous literature, which typically considers only the decomposition in Equation 7 (taking total uncertainty as the variance of the prediction), empirical results (see App E) demonstrate that using the target epistemic uncertainty $\mathcal{E}{\delta}$ is more effective than relying solely on the original epistemic uncertainty $\mathcal{E}$ from Equation 8.

---

> > ### Comment · Reviewer_7m4h · 2024-11-20
> > **More Concerns**
> >
> > Thank you for the author's explanation. However, I still have some concerns as follows:
> >
> > 1. I believe that Figure 1(c) needs further refinement. The absence of a legend makes it difficult to comprehend. Specifically, there is no explanation for the significance of the blue and green colors. Adding a legend and clarifying these colors would substantially improve the figure's clarity.
> >
> > 2. I still think it is important to apply UPER to some SOTA deep RL algorithms. Focusing solely on outperforming PER might narrow the scope and reduce the impact on the community. Demonstrating that UPER can enhance a range of SOTA deep RL algorithms would showcase its broader applicability and potential contribution to the field.

---

> > > ### Author Response · Authors · 2024-12-04
> > >
> > > Thanks for your response. (1) apologies, we have updated the figure legend to improve the clarity. (2) we appreciate this perspective, and have performed some additional experiments with a different model, C51 (please see general response), which is the best we could do given the time constraints. We hope that by evaluating on another RL model helps to demonstrate the robustness of UPER and showcase its broader applicability. Thanks again for your feedback and contribution to our paper.

---

> ### Author Response · Authors · 2024-11-19
>
> > Theoretical Analysis
>
> We agree that theoretical analysis could help substantiate and guarantee improvements to our method. While we explored this during development, it remains challenging due to the coupling between value estimation and buffer sampling. Existing work on replay buffer dynamics ([1] from reviewer C2gf) would require incorporating epistemic and aleatoric uncertainty, which is beyond our proposal's scope. Nonetheless, we demonstrate our method's effectiveness through toy models, simulations, and ablation studies. For instance, Appendix G.1 shows that performance improvements on the Atari-57 benchmark stem solely from the prioritization variable. Additionally, Appendix D provides insights into the effects of prioritization based on uncertainty.
>
> UPER is not an ad hoc solution as it was derived in a principled way. The epistemic and aleatoric uncertainty decomposition, derived from the expected error in Equation 9, along with the information gain, are both mathematically derived (see Appendices B and D.1), hence UPER is a general prioritization scheme, and its quality will depend on the method used to estimate the distribution.
>
> __Refs__
>
> Statistics and Samples in Distributional Reinforcement Learning (Rowland et. al 2019)

---

### Official Review · Reviewer_C2gf · 2024-11-03

**Soundness:** 3
**Presentation:** 3
**Contribution:** 2
**Rating:** 5
**Confidence:** 3

**Summary:**

The paper introduces an approach called Uncertainty Prioritized Experience Replay (UPER) to enhance sample efficiency in deep reinforcement learning. The traditional Prioritized Experience Replay (PER) relies on temporal difference (TD) error to prioritize samples in the replay buffer which does not perform well in noisy environments. UPER addresses this issue by utilizing epistemic uncertainty—representing uncertainty that can be reduced through learning—along with aleatoric uncertainty, which refers to the inherent randomness in the data. The authors use these uncertainties to rank samples in the replay buffer. The authors validate UPER through experiments conducted on toy models and the Atari-57 benchmark, demonstrating that it outperforms PER and vanilla QR-DQN.

**Strengths:**

1) Motivating examples that clearly show the advantage over PER.
2) The paper is well-written and clearly presented.

**Weaknesses:**

1) The method is only compared to Prioritized Experience Replay (PER), which is quite old. There are other approaches that have improved upon PER in recent years, such as [1]. We expected to see comparisons with these newer methods.

2) The paper claims that it enhances sample efficiency; however, it does not demonstrate how their approach contributes to solving one of the most challenging Atari games, Montezuma’s Revenge, in the main paper. An experiment in the appendix reveals no improvements for these difficult problems.

3) Both PER and UPER are more computationally expensive than Uniform Sampling. We anticipated a comparison between these algorithms using the same training time or, at the very least, a comparison based on wall-clock time.





[1]- Understanding and mitigating the limitations of prioritized experience replay.

**Questions:**

1) PER can be applied in all RL algorithms which are based on TD error. How can we apply your idea in algorithms such as vanilla DQN where we do not have an ensemble of models or the distribution of Q-values?

2) Can you explain how Equations 7 and 8 are connected? I mean if we sum both uncertainties in Equation 8, can we get the total uncertainty in Equation 7?

---

> ### Author Response · Authors · 2024-11-19
>
> Thank you for your review. We are pleased that you found the experiments to clearly demonstrate UPER's advantages and that you consider the paper well-written. You raise valid points regarding the comparison with [1], the performance of Montezuma’s Revenge, and the applicability of our proposed method to vanilla DQN. We are committed to addressing each of your concerns in detail, and we will incorporate these responses into the revised manuscript. Please find the details below:
>
> > W1: Comparison with Langevin Dynamics Sampling method [1]
>
> In the referenced study, the authors address limitations of PER that are unrelated to uncertainty. Specifically, they tackle two issues: (1) outdated priorities due to batch-like priority updates in the replay buffer, and (2) insufficient sample space coverage, especially in high-dimensional problems. To address these, they train a model based on a Langevin dynamical system to estimate state transitions. This model generates "ideal transitions" with updated priorities and better sample space coverage. We argue that the strengths of their method are complementary to uncertainty estimation. While our approach may face issues with outdated priorities and sample space coverage, their method will be subject to aleatoric uncertainty. Although integrating their advantages into our method seems like a promising research direction, this falls beyond the scope of our paper and could merit an entirely new study.
>
> > W2: Montezuma’s Revenge.
>
> As you correctly noted, Montezuma’s Revenge is a challenging game for RL agents due to its sparse rewards and the need for long-term strategies, requiring deep exploration of possible trajectories. Several approaches have been proposed to address this, such as imitation learning in “Playing Hard Exploration Games by Watching YouTube” (Aytar et al., 2018), promoting exploration in “Go-Explore” (Ecoffet et al., 2021), and assigning intrinsic rewards to uncertain states in “Flipping Coins to Estimate Pseudocounts for Exploration in Reinforcement Learning” (Lobel et al., 2023). These strategies generally involve updating policies with highly exploratory trajectories. While UPER affects the resulting policy after training, it does not explicitly promote exploration. Therefore, we do not expect our method to improve performance in Montezuma’s Revenge, as its primary goal is prioritizing memory buffer transitions based on uncertainty measures, not fostering exploration. However, as we mentioned earlier, our approach could complement exploration-focused methods, making this a promising direction for future research. We will include this important discussion in the revised manuscript.
>
> > W3: Computational cost of PER, UPER, and uniform sampling.
>
> PER and UPER are indeed more computationally expensive than uniform sampling, but the additional cost is minimal. For PER, the main extra calculations involve computing priority probabilities (Equation 2) and maintaining priorities for all transitions in the buffer. According to the original Prioritized Experience Replay paper (Schaul et al., 2016, Appendix B.2.1), the runtime overhead compared to uniform sampling is only 2%-4%, with negligible memory impact. Since we use DeepMind’s DQN Zoo implementation, our method incurs the same small overhead.
>
> > Q1: Can UPER be used in a Vanilla DQN?
>
> This is an excellent question, and the short answer is "probably not." Uncertainty measures are typically derived from higher-order moments of the underlying and estimated probability distribution. A standard DQN agent estimates only the mean and lacks a concept of how volatile this estimate might be. To capture such volatility, the agent must estimate moments beyond the mean.
>
> This is why many works extend the vanilla DQN architecture. For example:
> - Direct Epistemic Uncertainty Prediction (Section 2.3.3) trains a network to predict sample-specific errors.
> - Lobel et al. (2023) add a network component to estimate counts as a proxy for uncertainty.
> - Direct Variance Estimation (Appendix A.1) introduces a separate network to estimate return variance.
> - Bootstrapped DQN (Section 2.3.1) uses ensembles to approximate the posterior distribution and capture higher-order moments.
> - Quantile regression (Section 2.3.2) includes a parameter for estimating different parts of the distribution through quantiles.
> These examples highlight the need for additional mechanisms to estimate uncertainty in Q-learning. Thus, a standard DQN alone cannot provide precise measures of epistemic and aleatoric uncertainty.

---

> > ### Author Response · Authors · 2024-11-19
> >
> > > Q2: Epistemic and Aleatoric uncertainty as Total Uncertainty
> >
> > You are correct. Adding aleatoric and epistemic uncertainty yields the total uncertainty, as shown in Equations 7 and 8. This approach, derived in the original paper by Clements et al. (2020), requires an estimation of the underlying data distribution and posterior (e.g. ensembles) to decompose total uncertainty. We will include this derivation in the appendix of our revised manuscript, similar to the one in Appendix B (“Total Error Decomposition”). The broader concept was introduced in Direct Epistemic Uncertainty Prediction (Lahlou et al., 2020).

---

> > ### Comment · Reviewer_C2gf · 2024-12-03
> > **Response to Authors**
> >
> > I am glad to hear that my comments were considered helpful.
> >
> > However, I believe that other strategies for the replay buffer, such as [1, 2, 3], are more advantageous because they can be applied to a wide range of reinforcement learning (RL) methods. If your method necessitates a distributed implementation, it should demonstrate significant improvements compared to other methods. While I am not fully familiar with the current state-of-the-art (SOTA) for replay buffers, it would be beneficial for future submissions to justify the need for a distributed implementation by showcasing the improvements it offers compared to SOTA.
> >
> > Thank you once again for your response, and I apologize for my delayed reply.
> >
> > [1] A Deeper Look at Experience Replay.
> > [2] Hindsight Experience Replay
> > [3] Event Tables for Efficient Experience Replay

---

> > > ### Author Response · Authors · 2024-12-04
> > >
> > > Many thanks for your response. We’d like to gently push back on a couple of your assertions. Firstly, we consider UPER in some sense to consist of two distinct contributions: the criterion itself i.e. the information gain governing the trade-off between epistemic and aleatoric uncertainty, and the implementation that we propose around distributions. Of course the latter necessitates a distributions and ensembles, but the criterion itself does not—one could in principle also perform such estimates with other methods (direct variance estimates, coin flipping [Lobel et. al 2023] etc.). Second, we would argue that our method (especially when considering the first of the two components mentioned above) _can_ be applied to a wide range of reinforcement learning methods, one would just need to supplement the models with additional estimators. Virtually any method that computes uncertainties will require estimating higher moments. More broadly, we view this as a conceptually interesting method. Thinking about uncertainties in this way for replay has not been done before; it clearly works well empirically, but we also hope it will inspire further research on the topic both in the context of RL in machine learning and further afield e.g. in neuroscience and psychology—this is another reason we believe it merits consideration. Finally, we argue that requiring our method to be evaluated on multiple models, while not unfair _per se_, is not entirely justified; many papers evaluate their methods on one model baseline (e.g. even the hindsight experience replay paper you referenced, with the computational resources of OpenAI no less, evaluates only on DDPG for their main experiments).
> > >
> > > Despite our protestations, the extended rebuttal period did give us the necessary time to perform some additional experiments. We implement UPER on an ensemble version of C51—another distributional method—and evaluate it on Atari with strong results.
> > >
> > > We thank you for your continued engagement with our work, it is much appreciated. We hope our additional results help to convince you of the value of our method.

---

### Author Response · Authors · 2024-11-21
**General Response**

Thank you to all reviewers for their careful reading, insightful questions, and valuable suggestions. We have addressed all concerns, significantly improving the paper. Here are key points based on the feedback:

One aspect of our proposed method UPER  we could have been clearer about is that it is not an algorithm that enhances exploration explicitly. Traditional exploration algorithms use uncertainty measures to encourage exploration by rewarding agents for interacting with unvisited, uncertain, or high-error states. In contrast, UPER employs estimators for epistemic and aleatoric uncertainty to prioritize sampling transitions from the replay buffer based on information gain, which is then used to update value estimates. While these uncertainty measures are partly inspired by exploration-focused reinforcement learning literature, UPER applies them with a different goal. One implication of this for instance, is that we do not necessarily expect significant improvements in exploration-heavy games like Montezuma’s Revenge. We appreciate the reviewer’s feedback and have added clarifications in the discussion and introduction sections, with further details on uncertainty estimation methods provided in Appendix A.

UPER can integrate with other reinforcement learning features, as it modifies only the priority variable for sampling transitions from the buffer. For instance, it could be combined with intrinsic motivation or other exploration techniques. However, we anticipate less effective results when combining UPER with categorical distribution methods like C51, Rainbow, or BBF. These methods tend to overestimate variance, require manually defined distribution support, and often concentrate probability mass in longer state trajectories (see Fig 2 in https://arxiv.org/abs/1902.08102). Besides, QR-DQN (https://arxiv.org/pdf/1710.10044) consistently performs better across various scenarios, and building a QR-DQN version of Rainbow is out of the scope of our work. Our experiments, spanning toy models, Atari benchmarks, and ablation studies in the appendix, focus on isolating the effect of the prioritization variable to confirm the contribution of using epistemic and aleatoric uncertainty for replay prioritization.

The estimators in UPER are derived from the total uncertainty defined in Equation 9, which measures the expected prediction error across quantiles and ensembles. While prior work (Clements et al., 2020) used prediction variance as total uncertainty, our approach considers expected prediction error instead. This adds a term to the epistemic uncertainty, akin to the commonly used TD error, capturing both the deviation from the target and ensemble disagreement. Our extension, termed target epistemic uncertainty in Equation 9, is crucial for effectively prioritizing transitions in the replay buffer. Appendix D includes extensive ablation studies on this term and alternative prioritization methods.

We are eager to engage in further discussion and address any additional questions or feedback.

Thank you.

---

### Author Response · Authors · 2024-12-04
**Additional Atari Results for UPER**

We thank the reviewers for their time and effort evaluating our work. While we do not fully agree (for various reasons outlined in the individual discussions), we understand the reviewers’ contention that the value of our work would be greater if we can demonstrate the effectiveness of UPER on multiple models. To the best of our understanding this represents the only significant remaining concern among the reviewers. The extended rebuttal period has allowed us to implement and evaluate UPER on an additional distributional algorithm, C51. Our results robustly show that UPER outperforms PER (on 5/6 Atari games we tested, with equal performance on the sixth—over 2 seeds). Since we can no longer update the manuscript, we have added the plots to [this anonymous link](https://www.dropbox.com/scl/fi/ssvqvoh3j9294hncfkqn7/c51_uper.pdf?rlkey=m9o5mqy8iu6fy157ass6egn9s&dl=0)—we will format these plots properly and add them to future manuscript versions; note that not all runs were able to finish fully. Despite initial concerns about variance over-estimation in categorical representations, UPER still finds a good trade-off between epistemic and aleatoric uncertainty estimates for effective use as a prioritization criterion. We think these results add significant support for our proposed method and consequently improve our paper overall; we thank the reviewers for encouraging this direction. We hope the reviewers appreciate the considerable effort we have put in to address their concerns, especially in producing additional Atari results, and will re-consider their scores accordingly.

---

### Meta-Review · Area_Chair_FkfP · 2024-12-20

**Metareview:**

The paper studied uncertainty priorized experience replay where they use epstimic uncertainty to prioritize the samples in the replay buffer. The authors demonstrated the efficacy of their approach using QR-DQN and C51 as the underlying distributional RL optimizers.

The main weaknesses identified by reviewers are lack of sufficient comparison to prior works that use different strategies for prioritized experience replay. The reviewers also concerned about the approach's applicability to other RL algorithms.

**Additional Comments On Reviewer Discussion:**

The reviewers raised a concern about the lack of comparison to other more recent approaches on prioritized experience replay and the applicability of the proposed idea to other distributional RL algorithms. The reviewers thought the paper does not show convincing advantages of the proposed approach over other experience replay baselines. During the rebuttal period, the authors added additional experimental results, which demonstrate that their approach can also be applied to C51, another popular distributional RL algorithm. While the reviewers acknowledged the authors' effort during the rebuttal time, they were unconvinced due to the limited number of random seeds and the incomplete nature of the experiments. The reviewers encourage the authors to conduct more thorough experiments in future submissions, including additional comparisons to more recent prioritized experience replay baselines and experiments on applying their approach to other distributional RL optimizers. The authors also initially believed their approach wouldn't work well for algorithms like C51. However, their additional experiment results showed that it indeed worked pretty well for C51. The authors should consider explaining this phenomenon in their future submissions.

---

### Decision · Program_Chairs · 2025-01-22

Reject